# Rare sugar ʟ-sorbose exerts antitumor activity by impairing glucose metabolism

Hui-Lin Xu[1], Xiaoman Zhou [1], Shuai Chen[1], Si Xu[1], Zijie Li [1✉], Hideki Nakanishi [1✉] & Xiao-Dong Gao [1,2✉]

Rare sugars are monosaccharides with low natural abundance. They are structural isomers of dietary sugars, but hardly be metabolized. Here, we report that rare sugar ʟ-sorbose induces apoptosis in various cancer cells. As a C-3 epimer of ᴅ-fructose, ʟ-sorbose is internalized via the transporter GLUT5 and phosphorylated by ketohexokinase (KHK) to produce ʟ-sorbose-1-phosphate (S-1-P). Cellular S-1-P inactivates the glycolytic enzyme hexokinase resulting in attenuated glycolysis. Consequently, mitochondrial function is impaired and reactive oxygen species are produced. Moreover, ʟ-sorbose downregulates the transcription of KHK-A, a splicing variant of KHK. Since KHK-A is a positive inducer of antioxidation genes, the anti-oxidant defense mechanism in cancer cells can be attenuated by ʟ-sorbose-treatment. Thus, ʟ-sorbose performs multiple anticancer activities to induce cell apoptosis. In mouse xenograft models, ʟ-sorbose enhances the effect of tumor chemotherapy in combination with other anticancer drugs. These results demonstrate ʟ-sorbose as an attractive therapeutic reagent for cancer treatment.

[1] Key Laboratory of Carbohydrate Chemistry and Biotechnology, Ministry of Education, School of Biotechnology, Jiangnan University, Wuxi 214122, China. [2] State Key Laboratory of Biochemical Engineering, Institute of Process Engineering, Chinese Academy of Sciences, Beijing 100190, China. ✉email: lizijie@jiangnan.edu.cn; hideki@jiangnan.edu.cn; xdgao@ipe.ac.cn

Sugars, in particular glucose, serve as sources of energy and organic carbon in mammalian cells. In normal tissues, glucose is metabolized via glycolysis, the tricarboxylic acid (TCA) cycle, and oxidative phosphorylation. However, metabolic pathways are altered by cellular and environmental conditions. For example, many types of cancer cells are dependent on aerobic glycolysis in that glucose is metabolized via glycolysis and lactic acid fermentation; this phenomenon is known as the Warburg effect[1,2]. Remodeling of the metabolic pathway is required for cancer cells to maintain their rapid proliferation and viability[3]. Therefore, in cancer cells, rate-limiting enzymes for glycolysis, including hexokinase (HK), phosphofructokinase, and pyruvate kinase, are often upregulated[4]. Targeting the remodeling could be an attractive therapeutic strategy for cancer treatment.

Some sugars are known to modulate glycolytic processes and influence the proliferation of cancer cells. For example, mannose is known to inhibit the growth of cancer cells[5]. Mannose exhibits antiproliferative activity because mannose-6-phosphate, which is generated by HK, inhibits glycolytic enzymes including HK itself. Although mannose does not induce cancer cell death, it can enhance the effect of tumor chemotherapy[5]. In contrast to mannose, fructose is reported to activate the proliferation of cancer cells[6–8]. Fructose is a monosaccharide widely used as a sweetener. After internalization in cells via the glucose/fructose transporter GLUT2 or the fructose-specific transporter GLUT5, the monosaccharide is phosphorylated to produce fructose-1-phosphate (F-1-P)[9]. Accumulation of F-1-P leads to inactivation of pyruvate kinase M2 (PKM2). The inhibitory effect of F-1-P is most likely mediated by its direct binding to PKM2. In cells under hypoxic conditions, such as small intestinal epithelial cells and tumors, inactivation of PKM2 is beneficial for their proliferation. Indeed, a previous study showed that feeding a high-fructose diet in mice caused intestinal tumor progression[6].

Ketohexokinase (KHK) is the enzyme that produces F-1-P from fructose. Intriguingly, alternative splicing of the KHK gene can generate two distinctive functional isoforms termed KHK-A and KHK-C[10]. KHK-C has a much greater affinity for fructose than KHK-A; phosphorylation of fructose is primarily mediated by KHK-C[11]. KHK-A can mediate the phosphorylation of proteins besides sugars[12,13]. The selective autophagy receptor p62 is one of the substrates of the kinase. Phosphorylation of p62 at Ser28 results in the activation of nuclear factor erythroid 2–related factor 2 (Nrf2) via degradation of its inhibitor Kelch-like ECH-associated protein 1 (Keap1)[14]. Nrf2 is a transcription factor that induces antioxidant genes to counteract ROS. The expression of KHK-A is upregulated in various cancer cells[10,14,15]. Thus, KHK-A is involved in the proliferation of cancer cells irrespective of the activity to produce F-1-P. The function of KHK-A in normal tissues remains elusive.

Rare sugars are defined as monosaccharides and their derivatives are rarely found in nature. Recent progress in the large-scale production of rare sugars enables us to analyze their biological activities[16]. Previous reports have shown that rare sugars exhibit beneficial activities including anti-obesity, anti-diabetic, anti-pathogenic microorganism, and antitumor effects[17–20]. For example, D-allose exhibits antitumor activity, although its mechanism remains elusive[21–24]. Here, we report that another rare sugar, L-sorbose, uniquely induces apoptotic death in cancer cells. It enters cells primarily through GLUT5 and is then converted to S-1-P by KHK. S-1-P inhibits the activity of HK, which induces mitochondrial ROS production and apoptotic cell death. In addition, S-1-P downregulates the expression of KHK-A by modulating the splicing mechanism, which results in attenuation of the Nrf2 antioxidation pathway. This founding reveals another antitumor effect of L-sorbose. Furthermore, L-sorbose enhances the effect of tumor chemotherapy in combination with sorafenib in mouse xenograft models, which can largely decrease the dosage of sorafenib during the treatment.

## Results

**L-Sorbose triggers mitochondrial apoptosis.** The rare sugar D-allose was previously reported to target several cancer cells. To determine whether other rare sugars have potential in cancer therapy, a cell viability screening employing six rare sugars was performed on six cancer cell lines (Fig. 1a). From this screening, we found that L-sorbose could kill various cancer cell lines. In particular, the growth of the liver cancer cell lines Huh7 and HepG2 was significantly inhibited by treatment with 25 mM L-sorbose; however, their viabilities were not influenced by treatment with the same concentration of D-allose (Fig. 1a). L-Sorbose exhibited half-maximum inhibitory concentration (IC50) values of 33.82 mM (24 h), 27.32 mM (48 h), and 30.88 mM (72 h) in Huh7 cells and 27.68 mM (24 h), 34.89 mM (48 h) and 22.60 mM (72 h) in HepG2 cells (Fig. 1b). Furthermore, long-term colony formation assay showed that L-sorbose treatment impaired the proliferation of these two liver cancer cell lines (Fig. 1c).

To examine whether the cell death induced by L-sorbose is apoptosis, cells treated with L-sorbose were stained with annexin V and propidium iodide (PI), and the ratio of apoptotic cells was measured. The ratio of early stage apoptosis (annexin V-positive, PI-negative cells) and total apoptosis (both annexin V- and PI-positive cells) were then revealed to be increased in the cells treated with 25 mM and 50 mM L-sorbose (Fig. 1d). In addition, the levels of cleaved caspase 3 and the ratio of BAX/Bcl2, which are the indicators for the mitochondrial apoptosis, were found to be increased in Huh7 cells following the L-sorbose concentration depend way (Fig. 1e).

Mitochondrial apoptosis can be induced by mitochondrial dysfunction and reactive oxygen species (ROS) production[25,26]. Indeed, the mitochondrial membrane potential (MMP) decreased significantly when Huh7 and HepG2 cells were treated with 25 mM L-sorbose for 3 h (Fig. 2a, b). ROS levels were significantly elevated after the treatment with 25 mM L-sorbose for 6 h followed by a significantly decreased cell viability (Fig. 2c–e). To determine whether L-sorbose-induced apoptosis is attributable to ROS production, Huh7, and HepG2 cells were treated with the antioxidant N-acetyl-L-cysteine (NAC). As shown in Fig. 2f, l-sorbose-induced cell death was abrogated by NAC treatment. To further verify that L-sorbose treatment causes an increased ROS production in mitochondria, intracellular ROS were stained with dichlorodihydrofluorescein diacetate (DCFH-DA) in Huh7 cells incubated with L-sorbose for 6 h (Fig. 2g). Fluorescence-activated cell sorting (FACS) analysis showed that the levels of DCFH-DA staining were higher in L-sorbose-treated cells (Fig. 2h). These higher DCFH-DA staining were decreased in the presence of mitoquinone mesylate (mitoQ), which is a reducer for mitochondrial produced ROS[27] (Fig. 2h). Similar results can be also confirmed when the dihydrorhodamine 123 was directly used to monitor mitochondrial ROS (Supplementary Fig. 1). On the other hand, the biomass of mitochondria remained no changes after the treatment of L-sorbose (Fig. 2i). In sum, our results suggest that L-sorbose can promote mitochondrial ROS production in cancer cells, which induces apoptotic cell death.

**L-Sorbose enhances the anticancer effect of sorafenib.** Sorafenib is a multi-kinase inhibitor used to treat liver, kidney, and thyroid cancers[28,29]. The IC50 values of sorafenib in Huh7 and HepG2 cells were 8.275 μM and 6.731 μM, respectively (Fig. 3a). To examine whether L-sorbose could enhance the anticancer effect of sorafenib, we then treated these cells with 2 or 4 μM sorafenib in combination with the L-sorbose (12.5 or 25 mM). As shown in

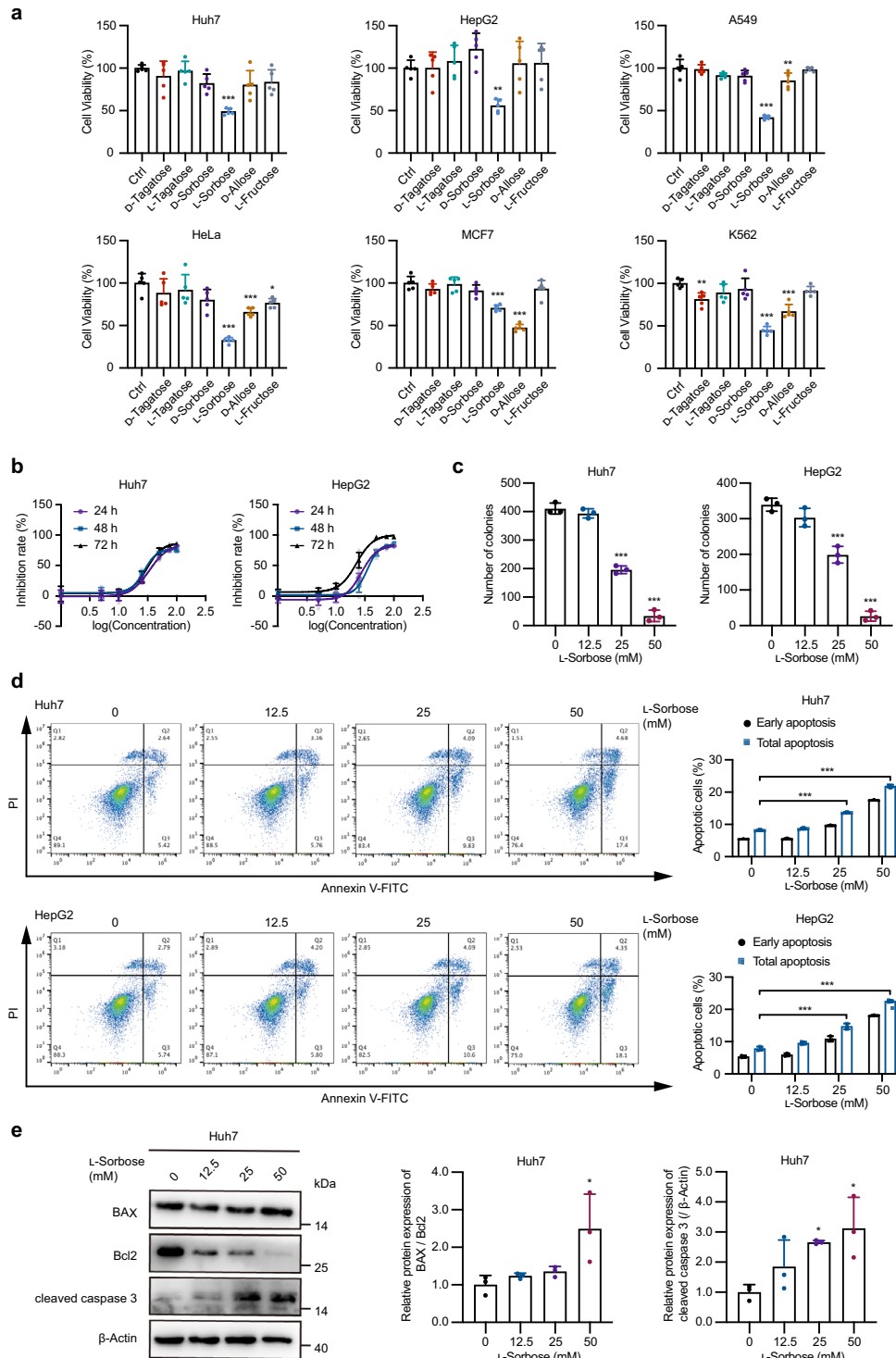

**Fig. 1 L-Sorbose induces cell apoptosis. a**, Cells were treated with or without (ctrl) 25 mM rare sugars for 48 h, and the viabilities were detected by CCK-8 assay. $n = 5$. **b**, Huh7 or HepG2 cells were incubated with various concentrations of L-sorbose for 24 h, 48 h and 72 h, and their viability rates were measured. $n = 5$. **c**, Cells were incubated in 0, 12.5, 25 or 50 mM L-sorbose. Long-term survival (14 days) was assayed by staining with crystal violet and the number of colonies was measured by ImageJ. $n = 3$. **d**, Cells treated with 0, 12.5, 25, or 50 mM L-sorbose for 24 h were stained with Annexin V-FITC and PI, and the ratios of apoptosis were measured by flow cytometry. Left panels: Representative dot plots. Right panels: Quantification of the ratio of early (annexin V-positive, PI-negative) and total (both annexin V- and PI-positive cells) apoptotic cells. $n = 3$. **e**, Left panels: Western blot showing the expression levels of BAX, Bcl2 and cleaved caspase 3 in Huh7 cells after 24 h with 0, 12.5, 25 or 50 mM L-sorbose treatment. Right panels: Quantification of the BAX/Bcl2 ratio and relative intensities of cleaved caspase 3. The BAX/Bcl2 ratio detected in Huh7 cells without L-sorbose treatment was defined as 1, and the relative ratios are shown. The intensity of cleaved caspase 3 normalized to β-actin detected in cells without L-sorbose treatment was defined as 1. $n = 3$. Data are presented as the mean ± s.d. and were analyzed by unpaired two-tailed Student's t test or one-way ANOVA with Dunnett's test. $*P < 0.05$, $**P < 0.01$, $***P < 0.001$.

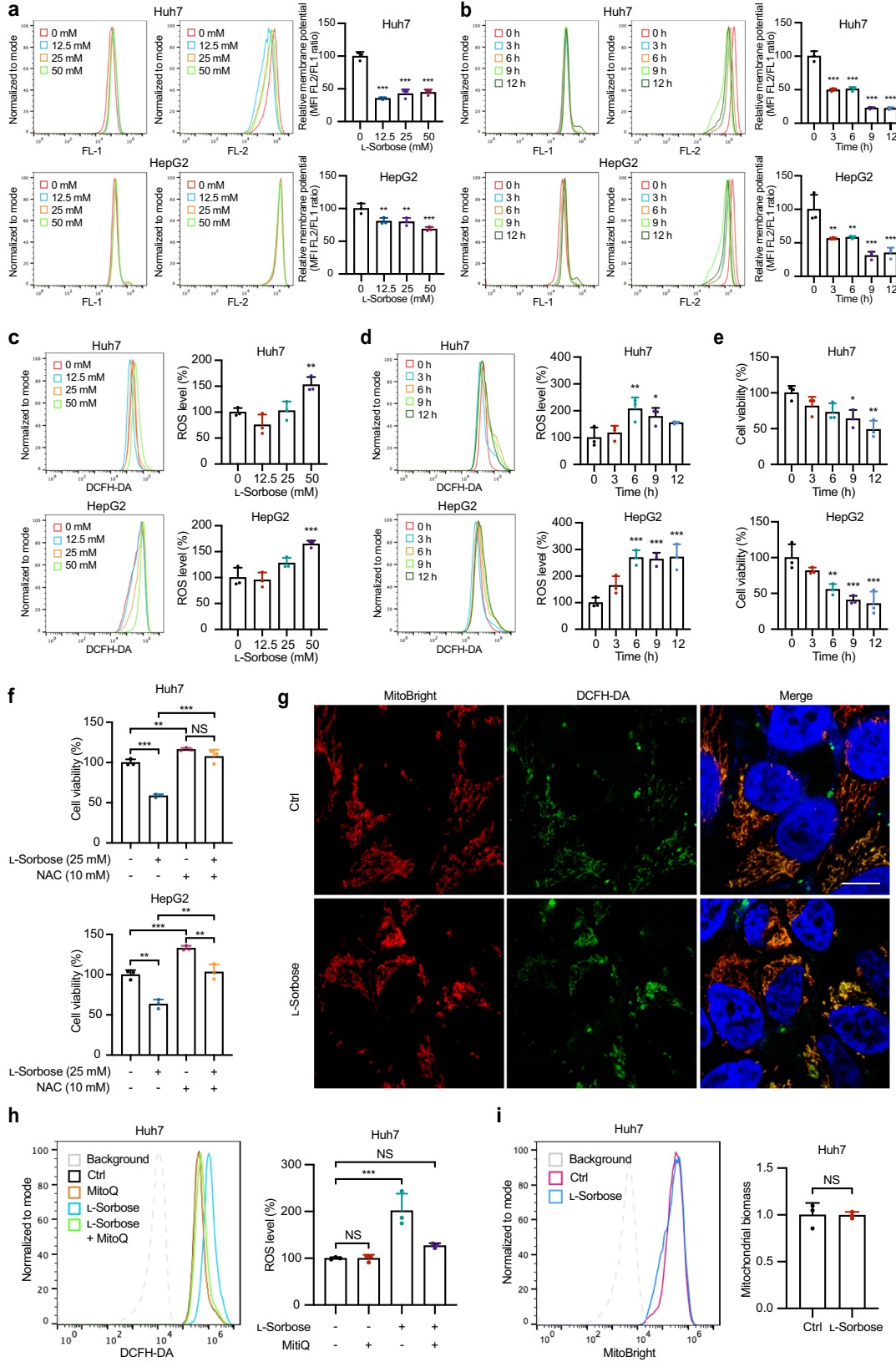

Fig. 3b, the combination of L-sorbose and low-dose sorafenib (2 μM) markedly decreased the cell viability compared with either alone. L-Sorbose- and sorafenib-treated cells died due to the induction of apoptosis (Fig. 3c). In line with this notion, this cell death was alleviated by incubation with the pancaspase inhibitor z-VAD-FMK (Fig. 3c, d), suggesting a synergistic anticancer effect between sorafenib and L-sorbose. This synergistic anticancer effect is L-sorbose specific because cell death was not observed when the cancer cells were incubated even with 25 mM of any other rare sugars and 2 μM sorafenib (Fig. 3e). To test

**Fig. 2 L-Sorbose triggers mitochondrial dysfunction and enhances ROS accumulation in cancer cells. a**, **b**, JC-1 dye was used to detect the mitochondrial membrane potential by flow cytometry. Left panels: Representative histograms. Right panels: Quantification of the FL2/FL1 MFI values. The data were normalized to the cells without L-sorbose treatment group (100%). Cells treated with 0, 12.5, 25, and 50 mM L-sorbose for 24 h (**a**). Cells treated with 25 mM L-sorbose at different time points (**b**). $n = 3$. **c**, **d**, The ROS level was detected by flow cytometry using DCFH-DA staining. Left panels: Representative histograms. Right panels: Quantification of the MFI values. The data were normalized to the cells without L-sorbose treatment group (100%). Cells treated with 0, 12.5, 25 or 50 mM L-sorbose treatment for 24 h (**c**). Cells treated with 25 mM L-sorbose at different time points (**d**). $n = 3$. **e**, Cell viabilities were analyzed after 25 mM L-sorbose treatment at different time points. $n = 3$. **f**, Cells pretreated with or without NAC (10 mM) for 30 min were incubated with vehicle or 25 mM L-sorbose for 24 h, and their viabilities were measured. $n = 3$. **g**, Representative images of Huh7 cells stained with DCFH-DA (green) and the mitochondrial marker MitoBright (red). Nuclei were stained with hoechst33342 (blue). Scale bar, 10 μm. **h**, Huh7 cells incubated with or without 500 nM mitoQ and 25 mM L-sorbose for 24 h were stained with DCFH-DA and subjected to FACS analysis. Left panels: Representative histograms. Right panels: Quantification of the MFI values. The data were normalized to the cells without L-sorbose and mitoQ treatment group (100%). $n = 3$. **i**, Cells were strained with a mitochondria marker, MitoBright, and analyzed by FACS. Left panels: Representative histograms. Right panels: Quantification of the MFI values. The MFI value detected in cells without L-sorbose treatment was defined as 1. $n = 3$. Data are presented as the mean ± s.d. and were analyzed by unpaired two-tailed Student's t test. $*P < 0.05$, $**P < 0.01$, $***P < 0.001$. NS, not significant.

whether L-sorbose share this synergistic effect with other chemotherapeutic drugs, we treated Huh7 and HepG2 cells with cisplatin (50 μM), doxorubicin (100 nM), lenvatinib (2 μM) and paclitaxel (1 μM) in combination with 25 μM of L-Sorbose. As shown in Fig. 3f, l-sorbose also exhibited a synergistic effect to kill cancer cells with these chemotherapeutic drugs.

To examine whether L-sorbose can enhance the therapeutic effect of sorafenib in vivo, nude mice subcutaneously injected with Huh7 cells were treated with L-sorbose and sorafenib either alone or in combination. The mice were given either normal drinking water or 20% L-sorbose by gavage for 4 weeks. Sorafenib was intragastrically administered to mice at a dose of 50 mg/kg every day from the eleventh day. We found that L-sorbose treatment resulted in growth inhibition of xenografts, as well as reductions in tumor weight. Noteworthy, a greater inhibitory effect on tumor growth was observed when sorafenib was administered in combination with L-sorbose in mouse xenograft models. After the trial, the tumor volume became less than half by the combined use of L-sorbose and sorafenib compared to L-sorbose or sorafenib treatment alone (Fig. 4a–c, Supplementary Fig. 2). In addition, the ratio of BAX/Bcl2 increased (Fig. 4d) and the ROS level was evaluated in the sorafenib combined with L-sorbose groups (Fig. 4e), while no notable body weight loss was observed in mice treated with L-sorbose and/or sorafenib (Fig. 4f). Furthermore, the glucose and insulin levels in plasma were in same levels, showing that L-sorbose does not impact mice glycemia levels (Fig. 4g, h).

**L-Sorbose internalizes cell through GLUT5, and accumulates as L-sorbose-1-phosphate.** Structurally, L-sorbose is the C-3 epimer of D-fructose[30] (Fig. 5a). Fructose is transported into cells mainly via GLUT5, the specific transporter for fructose. Upon entering the cells, fructose is phosphorylated by KHK to D-fructose-1-phosphate (F-1-P)[31] (Fig. 5a). We found that the addition of D-fructose alleviated the cytotoxicity of L-sorbose (Fig. 5b), indicating the possibility that L-sorbose and fructose may compete the same transporter for internalizing cell or the KHK for phosphorylation (Fig. 5a).

To test our hypothesis, we examined whether GLUT5 is required for the uptake of L-sorbose. *SLC2A5* (the gene name of GLUT5) was knocked out in Huh7 and HepG2 cells; loss of GLUT5 protein in *SLC2A5* KO cells was verified by western blotting (Fig. 5c). Our results confirmed that *SLC2A5* KO cells showed greater tolerance to L-sorbose even under 100 mM L-sorbose (Fig. 5d), suggesting that, similar with the D-fructose, the cellular uptake of L-sorbose is mediated by GLUT5.

After internalizing cells, monosaccharides are often phosphorylated in the first step of their metabolism. Accordingly, accumulation of L-sorbose-1-phosphate (s-1-P) was observed in

cells incubated with L-sorbose (Fig. 5e). Since ketohexokinase (KHK) can phosphorylate furanoses including fructose, we examined whether phosphorylation of L-sorbose was mediated by this kinase. KHK has two isoforms KHK-A and KHK-C; KHK-C exhibits greater activity to phosphorylate fructose[11]. Thus, a plasmid containing human KHK-C gene was transformed into Huh7 and HepG2 cells (Fig. 5f). It was observed that S-1-P level was increased in cells with overexpression of KHK-C (Fig. 5e). Furthermore, we performed an in vitro phosphorylation assay using a recombinant human KHK-C prepared from *E. coli* cells. This experiment provided direct evidence that KHK-C can catalyze the conversion of L-sorbose into S-1-P (Fig. 5g).

**The expression level of KHK dictates L-sorbose sensitivity.** We confirmed that, by overexpression of KHK-C, Huh7 and HepG2 cells became more sensitive to L-sorbose treatment and the levels of apoptotic cell death were elevated (Fig. 6a, b). In contrast, the knockout of KHK in these cells prevented the cell lines from being affected by L-sorbose cytotoxicity (Fig. 6c, d). These results revealed KHK as a critical enzyme to dictate sensitivity to L-sorbose.

To further support the above conclusion, we analyzed the expression level of endogenous KHK-C in various L-sorbose treated cancer cell lines and compared their cell viabilities (Fig. 6e, f). The results showed that the L-sorbose sensitivity of these cells depends on their KHK-C expression levels. For instance, under the treatment with 25 mM L-sorbose, A549 cells with the highest expression of KHK-C revealed <50% viability, while MCF7 and T24 cells were almost resistant to the L-sorbose treatment (Fig. 6e, f). We speculated that the resistance of these cells to is attributable to lower expression levels of KHK-C. In line with this hypothesis, we proved that overexpression of KHK-C in MCF7 and T24 cells is able to recover the sensitivity to L-sorbose, resulting in a decrease in their cell viability (Fig. 6g, i) and an increase in apoptosis levels as well (Fig. 6h). As another line of evidence, we found that expression level in mice tumor was increased in the L-sorbose treated group and the combination group, indicating the importance of KHK-C in L-sorbose treatment (Fig. 6j).

**L-Sorbose-1-phosphate interferes with glucose metabolism by targeting the hexokinase activity.** To gain insight into the molecular basis for the anticancer activity of L-sorbose, we performed central carbon metabolomic analyses. The results showed that treatment with L-sorbose in Huh7 cells led to marked decreases in several intermediate metabolites in glycolysis and the tricarboxylic acid (TCA) cycle, including glucose-6-phosphate (G-6-P), 3-phosphoglyceric acid, L-lactic acid, and α-ketoglutarate (α-KG) (Fig. 7a). The decrease in G-6-P levels was of particular

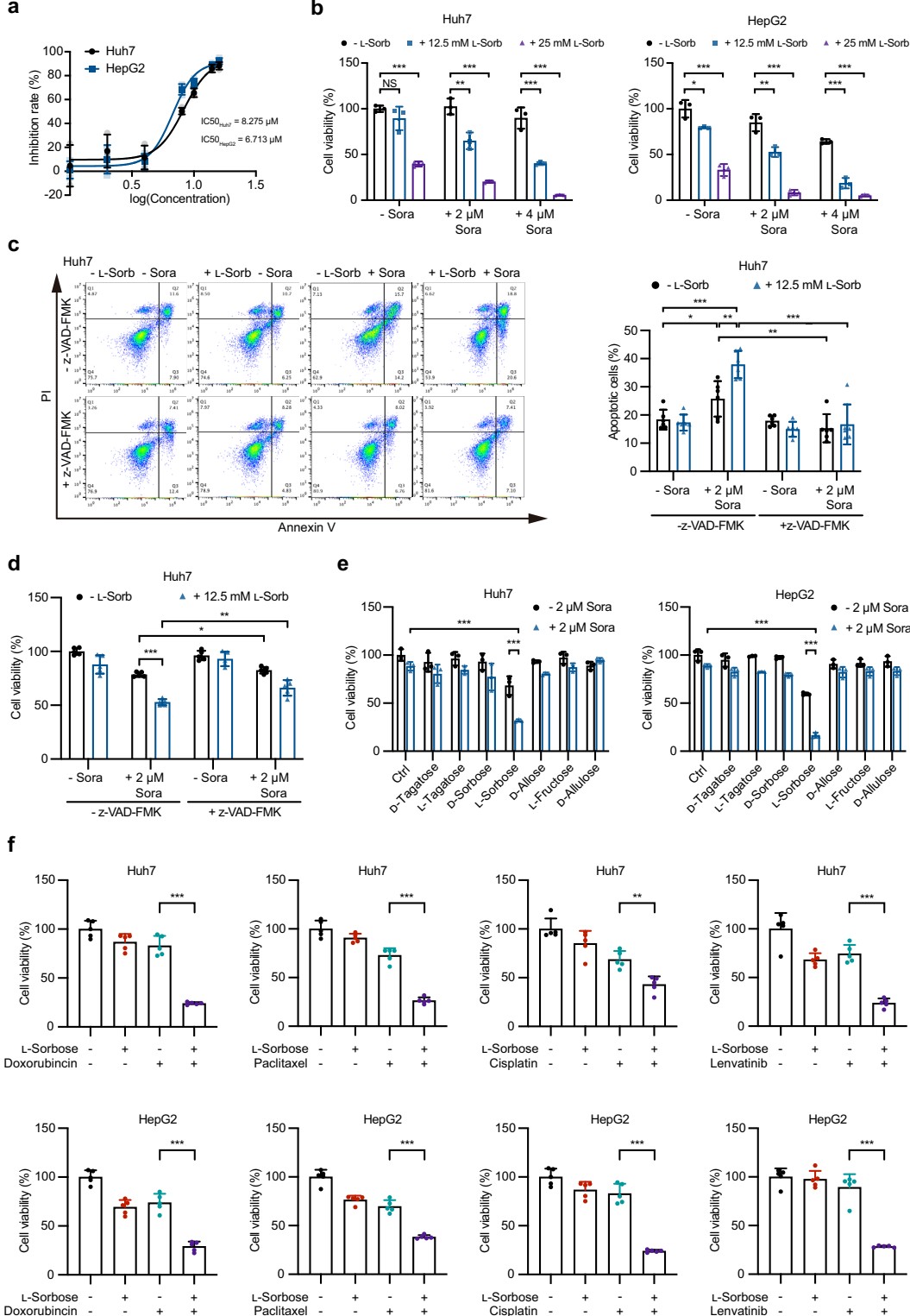

**Fig. 3 L-Sorbose enhances antitumor activity of sorafenib in vitro. a**, Huh7 or HepG2 cells were incubated with various concentrations of sorafenib for 24 h, and their viability rates were measured and IC50s were calculated. $n = 3$. **b**, Cells treated with L-sorbose (0, 12.5 or 25 mM) were incubated with sorafenib (0, 2 or 4 μM) for 24 h, and cell viability rates were measured. $n = 3$. **c**, Huh7 cells were treated with or without 12.5 mM L-sorbose, 2 μM sorafenib, and 50 μM z-VAD-FMK for 24 h. The levels of apoptosis were measured. Left panels: Representative dot plots. Right panels: Quantification of the ratio of apoptotic cells. $n = 6$. **d**, Huh7 cells were treated with or without 12.5 mM L-sorbose, 2 μM sorafenib, and 50 μM z-VAD-FMK for 24 h, and cell viability was measured. $n = 5$. **e**, Huh7 and HepG2 cells were treated by 2 μM sorafenib in combination with different rare sugars (25 mM) for 24 h, and their viabilities were measured. $n = 3$. **f**, Viabilities of cells treated with L-sorbose and other drugs. The concentrations of drugs used in this assay were as follows: cisplatin, 50 μM; doxorubicin, 100 nM; lenvatinib, 2 μM; paclitaxel, 1 μM. $n = 5$. Data are presented as the mean ± s.d. and were analyzed by unpaired two-tailed Student's t test or two-way ANOVA with Dunnett's test. *$P < 0.05$, **$P < 0.01$, ***$P < 0.001$. NS, not significant.

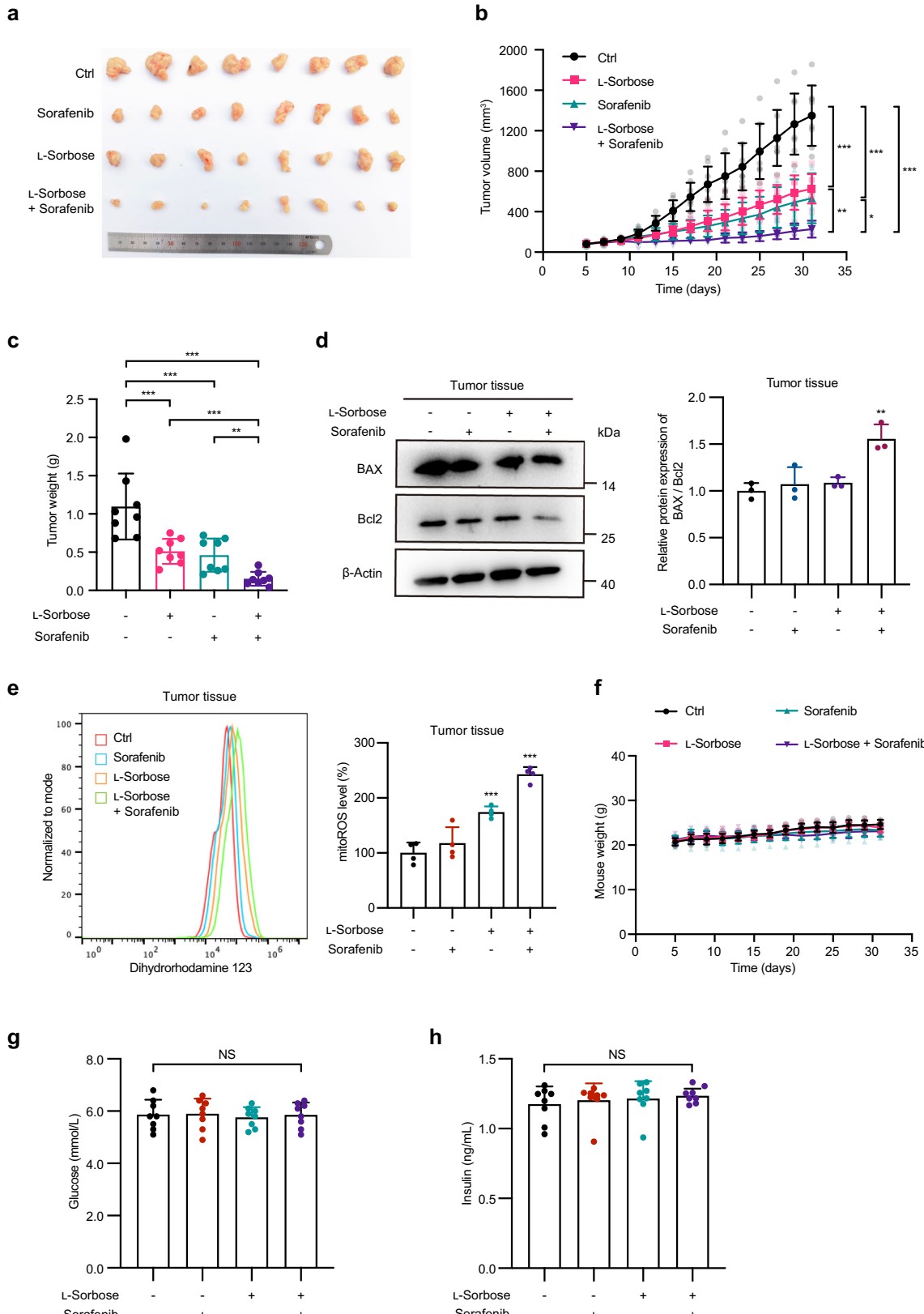

interest because hexokinase (HK), the enzyme required for converting glucose to G-6-P, is a viable target for cancer therapy. Then, we measured the metabolic flux of several intermediate metabolites in glycolysis. It was confirmed that the flux of the first metabolite of glycolysis G-6-P and the final metabolite of glycolysis lactic acid were decreased, indicating that L-sorbose

treatment interfered with the glycolysis in L-sorbose-treated Huh7 cells (Supplementary Fig. 3). Indeed, the kinase activity in lysates of Huh7 and HepG2 cells was decreased by L-sorbose treatment (Fig. 7b).

To further verify that the cytotoxic effect of L-sorbose is attributable to a decrease in HK activity, we assessed whether the

**Fig. 4 L-Sorbose enhances antitumor activity of sorafenib in vivo. a–c** BALB/c nude mice were injected subcutaneously with Huh7 cells and received either normal drinking water or 20% L-sorbose by oral gavage every day from the fifth day after tumor transplantation. Sorafenib was intragastrically administered to mice at a dose of 50 mg/kg every day from the eleventh day. The number of mice was n = 8 per group. Images show tumors of all mice (**a**). Tumor volume (**b**) and weight (**c**) were measured. **d**, Left panels: Western blot showing the expression levels of BAX and Bcl2 mouse tumor tissues. Right panels: Quantification of the Bax/Bcl2 ratio. The Bax/Bcl2 ratio detected in the group without L-sorbose and sorafenib administration was defined as 1. n = 3. **e**, The mitochondrial ROS level was detected by Dihydrorhodamine 123 in mouse tumors. Left panels: Representative histograms. Right panels: Quantification of the MFI values. The data were normalized to the group without L-sorbose and sorafenib administration (100%). n = 4. **f**, Mouse weight was measured every day from the fifth day after tumor transplantation. The number of mice was n = 8 per group. **g**, **h**, Fasting plasma glucose (**g**) and fasting plasma insulin (**h**) of tumor-bearing mice. n = 8. Data are presented as the mean ± s.d. and were analyzed by unpaired two-tailed Student's t test or two-way ANOVA with Dunnett's test. *P < 0.05, **P < 0.01, ***P < 0.001. NS not significant.

effect of L-sorbose was abolished by galactose because, unlike glucose and mannose, galactose enters glycolysis without using HK[32,33]. As shown in Fig. 7c, l-sorbose treatment resulted in substantial decreases in cell viability of Huh7 and HepG2 cells cultured with glucose or mannose; however, the cytotoxicity was significantly alleviated by culturing cells in glucose-free DMEM supplemented with galactose, and so does the ROS level (Fig. 7d). Meanwhile, the galactose itself does not influence the cell viability (Supplementary Fig. 4). Our results suggest that the apoptotic cell death induced by L-sorbose treatment is attributable to the glycolytic HK activity.

In many cancer cells, the Type II isoform of hexokinase (HK2) is overexpressed. We then characterized the HK2 protein in Huh7 and HepG2 cells. It was found that the expression, localization and phosphorylation of HK2 were not altered by L-sorbose treatment (Fig. 7e, f, Supplementary Fig. 5), excluding their involvement in reducing HK2 activity. We then incubated the recombinant human HK2 enzyme with different concentrations of S-1-P for confirming its inhibitory effect on HK activity. As shown in Fig. 7g, S-1-P inhibited HK2 activity with 4.35 mg/mL of IC50. These results demonstrated that the cytosolic S-1-P in cancer cells produced by L-sorbose treatment directly inhibits the HK2 activity.

**L-Sorbose treatment inactivates Nrf2-regulated antioxidant defense.** In cancer cells, KHK-A isoform is generally upregulated to support their survival by counteracting ROS production[14]. Intriguingly, we found that KHK-A levels were decreased in cancer cell lines Huh7 and HepG2 treated with L-sorbose, while KHK-C levels were elevated (Fig. 8a, b). In contrast, these effects were not observed in T24 cells, which is insensitive to L-sorbose (Fig. 8b), as well as in the Huh7 cells cultured with galactose supplemented DMEM (Supplementary Fig. 6). These observations suggested KHK-A as an alternative target of L-sorbose treatment.

Since KHK-A serves as a positive regulator to induce antioxidative genes via activation of Nrf2, a decrease in KHK-A levels may be another cause of apoptosis induction in L-sorbose-treated cancer cells. Thus, we assessed whether L-sorbose treatment affects the Nrf2 pathway. In cancer cell lines, Nrf2 is predominantly detected in the nuclear fraction. However, when the cells were treated with L-sorbose, the levels of Nrf2 detected in the nuclear fraction were significantly decreased (Fig. 8c). In addition, antioxidative genes regulated by Nrf2, including heme oxygenase-1 (HO-1), glutamate-cysteine ligase catalytic (GCLC), quinone oxidoreductase 1 (NQO1), and phosphogluconate dehydrogenase (PGD), were downregulated in L-sorbose-treated cells (Fig. 8d). Notably, KHK-A levels were not decreased by L-sorbose treatment in the normal hepatocyte cell line WRL68 (Fig. 8b), nor does the cell viability (Supplementary Fig. 7). Nrf2 levels detected in the nuclear fraction were not altered by L-sorbose treatment in WRL68 cells (Fig. 8c). These results suggest that L-sorbose modulates the splicing switch to generate

KHK isoforms in cancer cells, which helps to induce apoptosis. A previous study showed that mannose exhibits anticancer activity via HK inactivation[5]. However, KHK-A levels were not altered by mannose and neither KHK-C (Supplementary Fig. 8). Thus, modulation of KHK splicing would be a unique anticancer activity for L-sorbose.

## Discussion

L-Sorbose is the intermediate of a fermentation process for manufacturing vitamin C[34]. Although it is rare in nature, L-sorbose can be efficiently produced by the bioconversion from D-sorbitol in *Gluconobacter* or *Acetobacter*[35]. Here, we show that the rare sugar L-sorbose induces apoptotic cell death in cancer cells. Our results show that L-sorbose exhibits two anticancer activities: ROS production induced by HK inactivation and attenuation of a cellular antioxidant defense mechanism by KHK-A downregulation. Our results suggest that L-sorbose-mediated apoptotic cell death is induced by the synergy of these effects (Fig. 9).

In cancer cells, various metabolites, including glucose-6-phosphate (G-6-P), 3-phosphoglyceric acid, L-lactic acid, and α-ketoglutarate (α-KG), are decreased by L-sorbose treatment (Fig. 7a). This result suggests that a number of enzymes required to produce the metabolites are inactivated by L-sorbose treatment. Nevertheless, we conclude that the anticancer activity is primarily attributable to inactivation of HK because cell death induced by L-sorbose treatment was significantly reduced by culturing cells in galactose-containing media (Fig. 7d). Galactose is converted to G-6-P without using HK[32]. HK is a rate-limiting enzyme in glycolysis, and its activity is generally elevated to satisfy their high metabolic demands in cancer cells[36]. Depletion of glucose or inactivation of critical glycolytic enzymes in cancer cells is known to induce oxidative stress[37]. Thus, ROS production in L-sorbose-treated cells is most likely attributable to a decrease in HK activity and a subsequent reduction in glucose metabolism. In L-sorbose treated cells, levels of HK2 were not altered, while total activity of HK2 in the cells was decreased, indicating that enzymatic activity of HK is inhibited by L-sorbose treatment (Fig. 7b, g). KHK is required for L-sorbose to exhibit anticancer activity, suggesting that inhibition of HK is mediated by S-1-P or its derivatives. A previous report showed that S-1-P inhibited HK in dialyzed bovine brain extract[38]. Although the detailed mechanism for HK inhibition by S-1-P remains unknown, S-1-P could serve as a direct inhibitor of the kinase (Fig. 7g). Given that KHK is a pivotal enzyme for L-sorbose to perform its anticancer activity, the expression levels of this kinase could be a marker to indicate sensitivity to L-sorbose. In support of this hypothesis, L-sorbose exhibits cytotoxicity in a manner dependent on the levels of KHK expression (Fig. 6).

Similar to L-sorbose, mannose suppresses the proliferation of cancer cells because phosphorylated mannose inhibits HK activity[5]. However, unlike L-sorbose, mannose does not induce apoptotic cell death. L-sorbose exhibits more severe anticancer

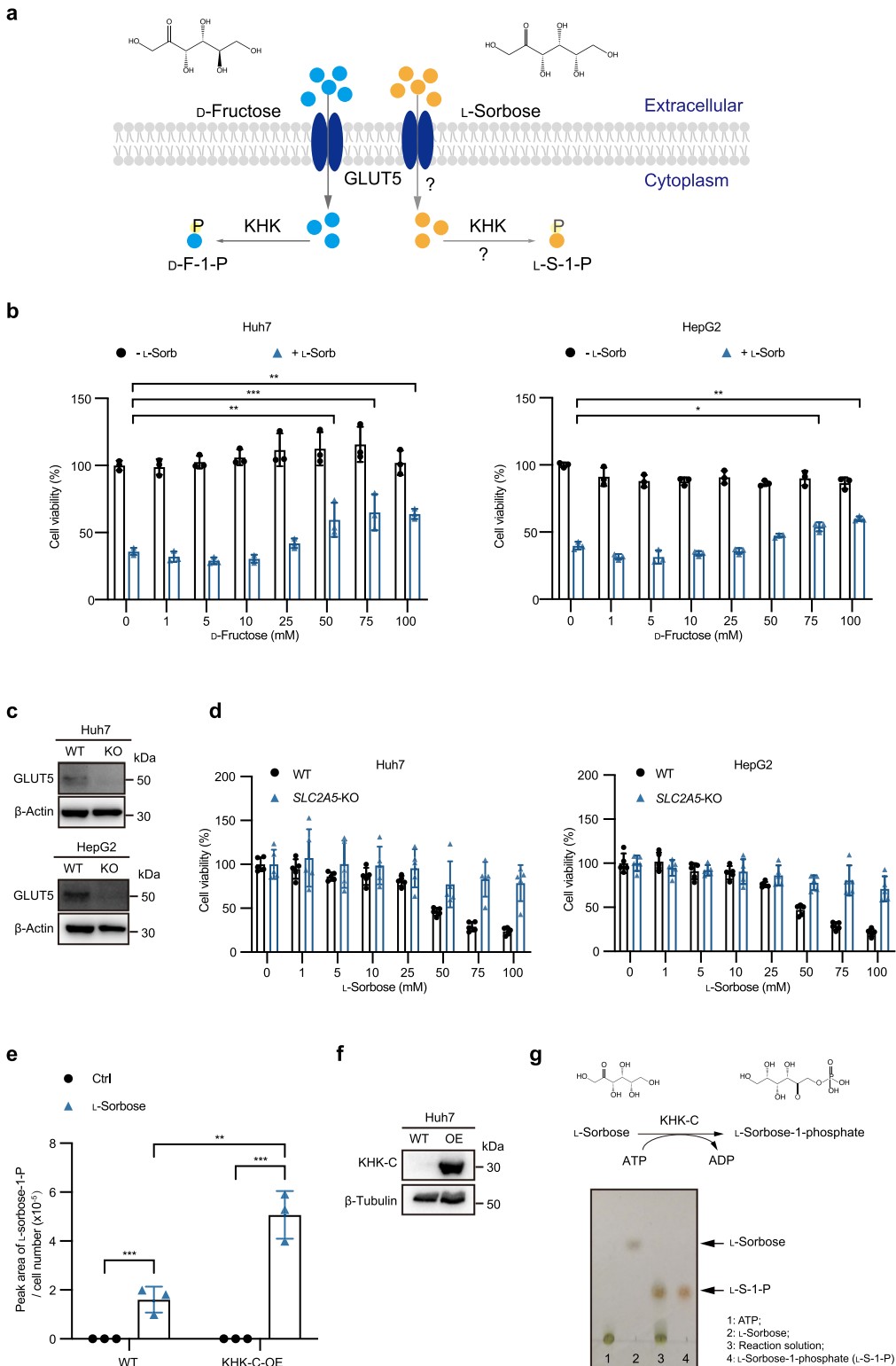

**Fig. 5 ʟ-Sorbose internalizes cells through GLUT5 and accumulates as ʟ-sorbose-1-phosphate. a**, Schematic diagram of first two steps of fructose metabolism and the hypothesis for ʟ-sorbose. **b**, Viabilities of Huh7 and HepG2 cells treated with or without ʟ-sorbose (25 mM) and the indicated concentrations of ᴅ-fructose for 24 h. $n = 3$. **c**, Western blot to detect GLUT5 in wild-type and *SLC2A5*-KO cells. **d**, Cell viabilities of wild-type or *SLC2A5*-KO cells treated with the indicated concentrations of ʟ-sorbose for 24 h. $n = 5$. **e**, Huh7 cells with or without overexpression of KHK-C were cultured in the presence or absence (ctrl) of ʟ-sorbose for 24 h. The intracellular amount of ʟ-sorbose-1-phosphate (ʟ-S-1-P) was detected by LC-MS. $n = 3$. **f**, Western blot verified the overexpression of KHK-C in Huh7 cells. **g**, Recombinant KHK-C was incubated with 200 μL 50 mM Tris–HCl buffer (pH 7.5) containing 5 g/L ʟ-sorbose, 3 mM MnCl₂, 3 mM MgCl₂ and 25 mM ATP. Top panels: The KHK-C reaction equation. Bottom panels: TLC analysis was performed to measure the production of ʟ-S-1-P. Data are presented as the mean ± s.d. and were analyzed by unpaired two-tailed Student's t test. $*P < 0.05$, $**P < 0.01$, $***P < 0.001$.

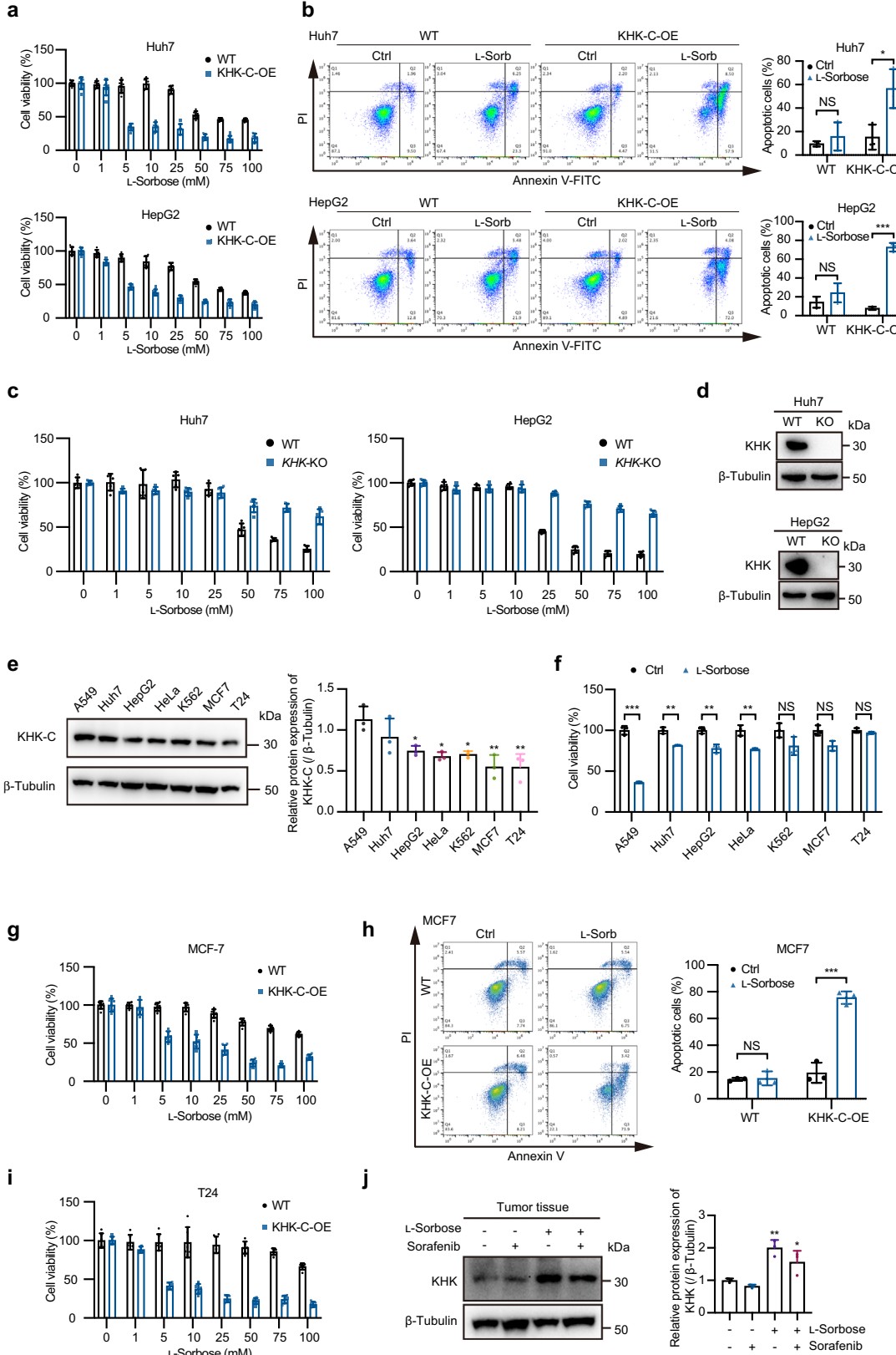

activity than mannose, presumably because the rare sugar performs an additional function; that is, conversion of KHK isoforms. Incubation of cancer cells with L-sorbose results in elevation of KHK-C levels (Fig. 8a, b). Genes involved in fructose metabolism are known to be induced by the uptake of fructose. Thus, like fructose, L-sorbose may be able to induce fructolytic

genes. In contrast to KHK-C, KHK-A is significantly decreased by the L-sorbose treatment (Fig. 8a, b). Downregulation of KHK-A causes attenuation of the Nrf2 antioxidation pathway, which should make cancer cells sensitive to ROS. Given that KHK-C and -A are generated via alternative splicing, these results indicate that the regulation of KHK splicing is modulated by L-sorbose

**Fig. 6 The expression level of KHK dictates L-sorbose sensitivity. a**, Cells with or without overexpression of KHK-C were treated with the indicated concentration of L-sorbose for 24 h and their viabilities were assayed. $n = 5$. **b**, Cells with or without overexpression of KHK-C were treated with 25 mM L-sorbose for 24 h and their apoptotic levels were measured. Left panels: Representative dot plots. Right panels: Quantification of the ratio of apoptotic cells. $n = 3$. **c**, Viabilities of wild-type and *KHK*-KO cells treated with the indicated concentrations of L-sorbose for 24 h. $n = 5$. **d**, Western blot to detect KHK in wild-type and *KHK*-KO cells. **e**, Left panels: Western blot of the endogenous KHK-C in different cancer cells. Right panels: Relative intensities of KHK-C. The intensity of KHK-C normalized to β-tubulin detected in A549 cells was defined as 1. $n = 3$. **f**, Cell viabilities on different cancer cells after 25 mM L-sorbose treatment or not for 12 h. $n = 3$. **g**, MCF7 cells with or without overexpression of KHK-C were incubated with the indicated concentration of L-sorbose for 24 h, and their viabilities were measured. $n = 5$. **h**, MCF7 cells with or without overexpression of KHK-C were incubated with the indicated concentration of L-sorbose for 24 h, and their apoptotic levels were measured. Left panels: Representative dot plots. Right panels: Quantification of the ratio of apoptotic cells. $n = 3$. **i**, T24 cells with or without overexpression of KHK-C were incubated with the indicated concentration of L-sorbose for 24 h, and their viability was measured. $n = 5$. **j**, Left panels: Western blot showing the expression levels of KHK in mouse tumor tissues. Right panels: Quantification of relative intensities of KHK. The intensity of KHK normalized to β-tubulin detected in the group without L-sorbose and sorafenib administration was defined as 1. $n = 3$. Data are presented as the mean ± s.d. and were analyzed by unpaired two-tailed Student's t test. *$P < 0.05$, **$P < 0.01$, ***$P < 0.001$. NS, not significant.

treatment. c-Myc, and heterogeneous nuclear ribonucleoproteins H1 and H2 are known to be involved in the splicing switch from KHK-C to KHK-A; thus, this pathway may be affected in L-sorbose-treated cells[10]. It is well known that some tumors prefer using other nutrition like glutamine or lactate to support their quick growth[39–41]. Even in those cancers, the effect of L-sorbose on the antioxidation pathway is still effective, revealing its wild application in cancer therapy.

We demonstrate that L-sorbose alone or in combination with other chemotherapy reagents exhibits anticancer activity in vitro and in vivo (Figs. 3, 4). It is notable that L-sorbose is reported to induce hemolysis in dogs due to inactivation of glycolysis in erythrocytes[42–44]. However, in erythrocytes derived from other animals, including humans, hemolysis is not induced by inactivation of glycolysis[42]. Indeed, no obvious health impact was observed in our mouse experiments. The toxicity of L-sorbose for humans has not been reported thus far. Thus, although further examinations are required regarding the safety and therapeutic effects of L-sorbose in vivo, the rare sugar would be an attractive reagent used for anticancer treatment.

## Methods

**Plasmids**. The oligonucleotides and plasmids used in this study are listed in Supplementary Table 1 and Supplementary Table 2. For gene overexpression, human *KHK-C* was cloned into pLVX-puro. For gene knockout, guide RNA sequences of *KHK* or *SLC2A5* were cloned into the pLVX-CRISPR-v2-puro vector. Human *KHK* was cloned into the pET28a vector for recombinant KHK expression in *E. coli*. All plasmids were sequenced and verified.

**Cell lines and cell culture conditions**. The cell line WRL68 was purchased from Mingzhou Biotechnology. Other cell lines were obtained from the Cell Bank of the Type Culture Collection of Chinese Academy of Sciences. Huh7, HepG2, A549, HeLa, K562, MCF7 and HEK293T cells were cultured in DMEM high glucose medium (Biological Industries, 0023119), and T24 and WRL68 cells were cultured in RPMI 1640 medium (Biological Industries, 2035127) with 10% fetal bovine serum (FBS) (Biological Industries, 1841924) in 5% CO₂ at 37 °C. All cells were identified without mycoplasma infection.

To overexpress genes in cultured cells, expression plasmids were stably transfected with a retrovirus-mediated transfection method. For this, HEK293T cells were transfected with pLVX-puro (empty plasmid) or pLVX-*KHK-C*-puro and the two packaging plasmids, psPAX.2 and pMD2.G. Two days later, the virus particles were collected, filtered, and used to infect the target cells. After 48 h of infection, transfected cells were selected with 2 μg/mL puromycin. Cells stably transfected with pLVX-puro were used as a control cell line.

**CRISPR–Cas9-mediated gene knockout**. HEK293T cells were transfected with pLVX-CRISPR-v2-puro (empty plasmid), pLVX-CRISPR-v2-*SLC2A5*-puro or pLVX-CRISPR-v2-*KHK*-puro and the two packaging plasmids, psPAX.2 and pMD2.G. Two days later, the virus particles were collected, filtered, and used to infect the target cells. After 48 h of infection, 2 μg/mL puromycin (InvivoGen, QLL-41-03) was used to select positive cells. The selected cells were subjected to limiting dilution to obtain knockout cells. Western blotting was used to verify the expression of the target genes. Selected knockout clones were analyzed and verified by DNA sequencing.

**Metabolomic analysis**. For central carbon metabolomic analyses, cells were seeded on 10 cm culture plates and cultured to a density sufficient for 70–80% confluence. After L-sorbose treatment for 12 h, the cells were collected and added to 400 μL water. Cells were vortexed for 1 min after the addition of 500 μL precooled chloroform/methanol (1/3, v/v) and then homogenized for 4 min at 35 Hz and sonicated for 10 min. The homogenization and sonication circles were repeated three times, vibrating at 4 °C for 15 min, and incubating at −80 °C for 1 h. The samples were centrifuged at $9000 \times g$ for 10 min. The supernatants were collected and dried under a gentle stream of nitrogen. Then, they were dissolved in 200 μL ultrapure water. Reconstituted samples were vortexed before filtration through a centrifuge tube filter and subjected to HPIC-MS/MS analysis.

The HPIC separation was performed using a Thermo Scientific Dionex ICS-6000 HPIC System (Thermo Scientific) equipped with Dionex IonPac AS11-HC (2 × 250 mm) and AG11-HC (2 mm × 50 mm) columns. Mobile phase A was 100 mM NaOH in water, and mobile phases C and D were methanol and water, respectively. Another pumping system was used to supply the solvent (2 mM acetic acid in methanol), and the solvent was mixed with the effluent before entering electrospray ionization (ESI) (flow rate of 0.15 mL/min). The column temperature was 30 °C. The temperature of the autosampler was 4 °C and the injection volume was 5 μL. An AB SCIEX 6500 QTRAP + triple quadrupole mass spectrometer (AB Sciex) equipped with an ESI interface was used for analysis and development. Typical ion source parameters were as follows: ion spray voltage = −4500 V, curtain gas = 30 psi, ion source gas 1 = 45 psi, ion source gas 2 = 45 psi, and temperature = 450 °C. By injecting a standard solution of a single analyte into the API source of the mass spectrometer, flow injection analysis was used to optimize the MRM parameters for each target analyte. AB SCIEX Analyst Workstation Software (1.6.3 AB SCIEX), MultiQuant 3.0.3. software and Chromeleon 7 were employed for MRM data collection and processing.

**Quantitative analysis of intracellular L-sorbose-1-phosphate**. Cells were seeded on 35 mm culture plates and cultured to a density of ~70%. After L-sorbose treatment for 24 h, the medium was aspirated, and 1 mL of 80% methanol:20% water mixture was added to extract metabolites. The plate was placed at −20 degrees for 20 min. Then, the cell material was scraped into a 1.5 mL test tube prechilled on ice. The cell debris was pelleted by centrifugation at $15,000 \times g$ at 4 °C for 10 min, and the supernatant was transferred to a new tube and stored at −20 °C until analysis.

HPLC separation was carried out using a WATERS ACQUITY UPLC System (WATERS) equipped with HILIC-Z 2.7 μm (2.1 × 100 mm) columns. The mobile phases were acetonitrile and 0.1% formic acid. The column temperature was 45 °C. The injection volume was 5 μL. The mass spectrometric data were collected on a MALDI SYNAPT Q-TOF mass spectrometer (WATERS) connected with an electrospray ionization interface in positive ion mode (ESI+). Typical ion source parameters were as follows: capillary voltage of 3.5 kV, cone voltage of 30 V, source block temperature of 100 °C, desolvation temperature of 400 °C, desolvation gas flow of 700 l/h, cone gas flow of 50 l/h, and collision energy of 6/20 eV. The mass range of m/z 20–2000 was scanned. MassLynx V4.2 software (WATERS) was employed for data acquisition and processing.

**Metabolic flux analysis using ¹³C₆-glucose**. Cells were seeded on 10 cm culture plates. The day after seeding, cells were washed three times with D-PBS before adding glucose-free DMEM (supplemented with 10% FBS, 4 mM glutamine, and 4.5 g/L ¹³C₆-glucose). After L-sorbose treatment for 12 h, the cells were collected and added 500 μL of MeOH/H₂O (3/1, v/v). Vortexed for 30 s and precooled in dry ice, repeated freeze-thaw three times in liquid nitrogen. The samples were vortexed for 30 s and sonicated for 15 min in the ice-water bath. Followed by incubation at −40 °C for one hour and centrifugation at $13,800 \times g$ and 4 °C for 15 min. A 400 μL aliquot of the clear supernatant was collected and dried by spin. Then the residue was reconstitution with ultrapure water according to the cell count. Reconstituted

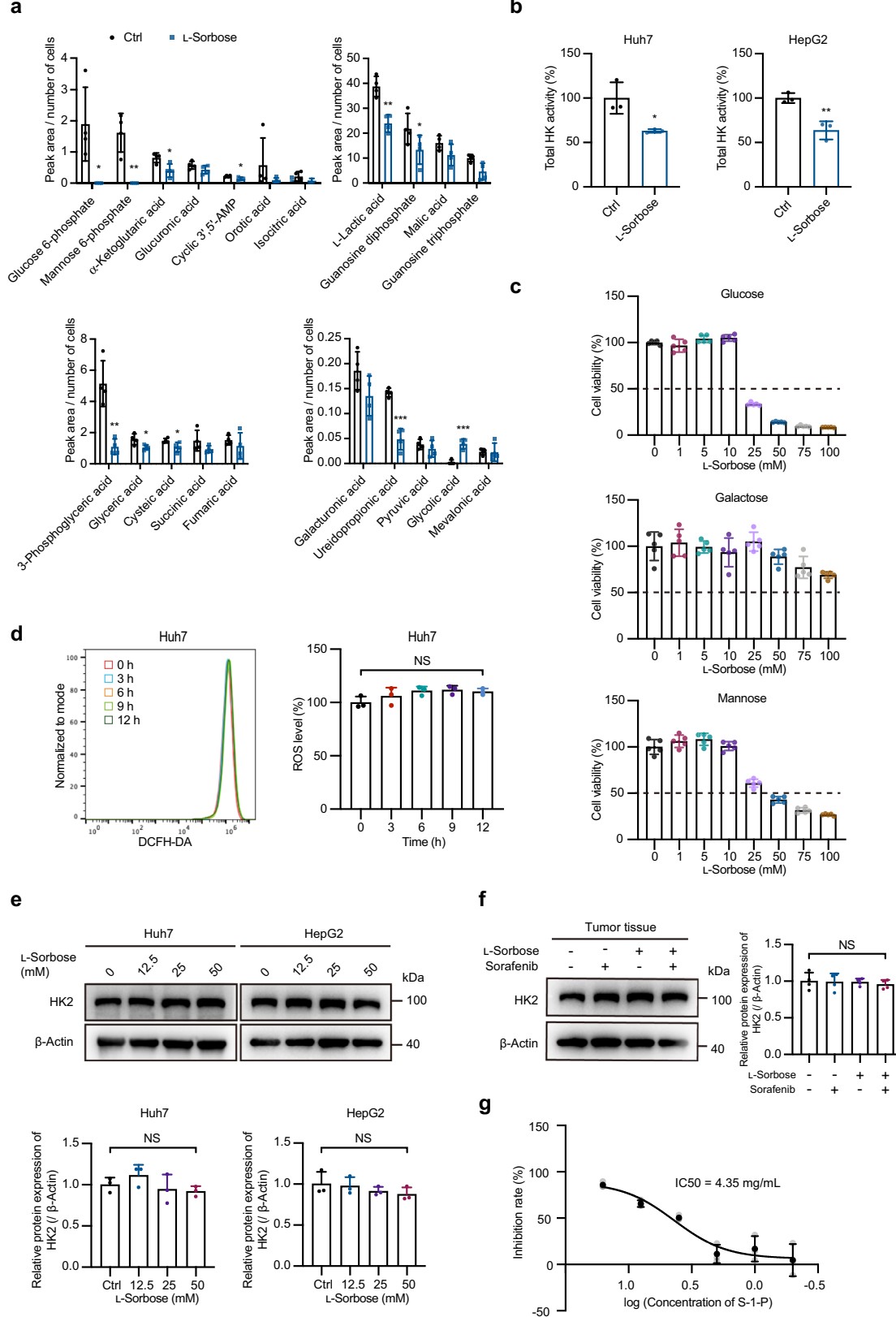

samples were vortexed before filtration through the centrifuge tube filter, and were subsequently transferred to inserts in injection vials for HPIC-QE-MS analysis after centrifugation at 13,800 × g and 4 °C for 15 min.

The HPIC separation was carried out using a Thermo Scientific Dionex ICS-6000 HPIC System (Thermo Scientific), equipped with Dionex IonPac AS11-HC (2 × 250 mm) and AG11-HC (2 mm × 50 mm) columns. The mobile phase A was 100 mM NaOH, and D was water, respectively. Another pumping system was used to supply the solvent (2 mM acetic acid in methanol) and solvent mixed with effluent before entering the ion source (flow rate of 0.15 mL/min). The column temperature was set at 30 °C. The auto-sampler temperature was set at 4 °C and the injection volume was 5 μL. The QE mass spectrometer was used for its ability to acquire MS spectra in full ms mode in the control of the acquisition software (Xcalibur 4.0.27, Thermo). In this mode, the acquisition software continuously evaluates the full scan MS spectrum. The ESI source conditions were set as follows:

**Fig. 7 S-1-P interferes with glucose metabolism by targeting the activity of hexokinase. a**, Differential metabolite analysis using central carbon metabolomics on Huh7 cells treated with or without (ctrl) 25 mM L-sorbose for 12 h. $n = 4$. **b**, Total activity of HK in cells treated with or without (ctrl) 25 mM L-sorbose for 12 h. $n = 3$. **c**, Cell viabilities on Huh7 cells cultured in glucose-free DMEM supplemented with the indicated sugars after L-sorbose treatment for 24 h. $n = 5$. **d**, The ROS level was detected by DCFH-DA in cells cultured in glucose-free DMEM supplemented with D-galactose after 25 mM L-sorbose treatment at different time points. Left panels: Representative histograms. Right panels: Quantification of the MFI values. The data were normalized to the cells without L-sorbose treatment group (100%). $n = 3$. **e**, Top panels: Western blot to analyze HK2 protein expression levels in cells. Bottom panels: Relative intensities of HK2. The intensity of HK2 normalized to β-actin. $n = 3$. **f**, Left panels: Western blot to analyze HK2 protein expression levels in mice tumor tissues. Right panels: Relative intensities of HK2. The intensity of HK2 normalized to β-actin. $n = 4$. **g**, Dose response inhibition of HK2 activity by S-1-P. $n = 3$. Data are presented as the mean ± s.d. and were analyzed by unpaired two-tailed Student's t test. *$P < 0.05$, **$P < 0.01$, ***$P < 0.001$. NS, not significant.

sheath gas flow rate as 30 Arb, Aux gas flow rate as 10 Arb, capillary temperature 350 °C, full MS resolution as 70,000, spray Voltage as −3.8 kV (negative). The raw data were converted to the mzXML format using ProteoWizard (Massconvert) and processed with an in-house program.

**Xenograft model**. All procedures were conducted in compliance with all the relevant ethical regulations and were approved by the Jiangnan University Animal Welfare and Ethical Review Body. Male BALB/c nude mice aged 5 weeks were purchased from Charles River Laboratories and placed in the Animal Experiment Center of Jiangnan University. Mice were placed five per cage with free access to water and food (chow diet). After habituation for a week, mice were inoculated subcutaneously with Huh7 cells ($1.0 \times 10^7$ cells in 100 μL PBS per mouse). When the tumors reached 50–100 mm³, the mice were randomly assigned to different groups. Sorafenib was dissolved in dimethyl sulfoxide (DMSO) and diluted in 5% sodium carboxymethyl cellulose. Mice received 20% L-sorbose in water according to their body weights (200 μL/20 mg) by oral gavage once every day. Sorafenib was intragastrically administered to mice at a dose of 50 mg/kg every day. Mice were sacrificed when the xenograft tumor size reached 1000 mm³. None of the mice showed severe weight loss or signs of infection or wounds. The tumor volume was measured every other day until the endpoint and calculated according to the equation: Volume = Length × Width² × 1/2.

**Measurement of mouse plasma glucose and insulin levels**. At the end of the xenograft model experiment, tail blood samples were collected after 10 h of food restriction and used for the measurement of plasma glucose using a glucometer (yuwell) and plasma insulin with a mouse insulin ELISA kit (Sangon Biotech, D721197).

**Immunohistochemistry**. Mice tumor tissues were fixed in 4% paraformaldehyde overnight, then dehydrated in ethanol and embedded in paraffin. IHC was performed using the anti-HK2 antibody (Proteintech, 22029-1-AP, 1:200). Images were obtained with a Nikon C2 Eclipse Ti-E microscope equipped with NIS-Element AR software and analyzed by FlowJo software.

**Western blot**. Cells were lysed in RIPA buffer (Solarbio, R0020) supplemented with protease inhibitor cocktail (MCE, HY-K0010) on ice for 30 min. After centrifugation at $15,000 \times g$ for 10 min at 4 °C, the protein lysates were collected. Then the protein lysates were separated by SDS–PAGE and transferred onto PVDF membranes. The membranes were blocked with 5% nonfat powdered milk for 1 h and incubated with primary antibodies against β-actin (Proteintech, 23660-1-AP, 1:1000), BAX (CST, 2772 T, 1:1000), Bcl-2 (CST, 3498 T, 1:1000), β-tubulin (Proteintech, 10068-1-AP, 1:1000), PCNA (BBI, D220014, 1:1000), Nrf2 (Abcam, ab137550, 1:1000), HK2 (Proteintech, 66974-1-lg, 1:5000), p-Tyr (CST, 9411, 1:1000), GLUT5 (Santa Cruz, sc-271005, 1:1000), KHK (Santa Cruz, sc-377411, 1:1000), KHK-A (SAB, 21709, 1:500), and KHK-C (SAB, 21708, 1:500) at 4 °C overnight and then probed with the appropriate secondary antibodies for 1 h at room temperature. The bands were visualized with western ELC substrate (BIO-RAD, 1705060), and the images were captured on a Tanon-5200Multi visualization instrument.

**Quantitative real-time PCR**. Total RNA was isolated from cells using CellAmp™ Direct RNA Prep Kit for RT–PCR (TaKaRa, 3732), and complementary DNA was synthesized from 5 μg total RNA using PrimeScript™ RT Master Mix (TaKaRa, RR036A). qPCR was performed using TB Green® Premix Ex Taq™ II (Tli RNaseH Plus) (TaKaRa, RR420A) on a Prism 7000 Sequence Detection System (Applied Biosystems) according to the manufacturer's instructions. Primer sequences for qPCR are listed in Supplementary Table 3.

**Cell viability assay**. Cells were seeded in 96-well culture plates at a density of $5.0 \times 10^3$ cells/well and grown overnight. After treatment with L-sorbose or other reagents, the cells were incubated with CCK-8 (Cell Counting Kit-8, DOJINDO

Laboratories, CK04) solution and cultured at 37 °C for another 1 h. The absorbance was detected at 450 nm wavelength using a microplate reader (BIO-RAD).

**Colony formation assay**. Cells were seeded in six-well plates at a density of 1000 cells/well. Cells were incubated for 24 h to allow attachment to the plates, after which L-sorbose was added to the cells and incubated for 24 h. The cells were cultured for 14 days in the absence of L-sorbose and fixed with a 4% paraformaldehyde fix solution (Beyotime Biotechnology, P0099) for 20 min. Then, the cells were stained with crystal violet (Beyotime Biotechnology, C0121) solution diluted in water. After 10 min, the plate was washed with water left to dry and scanned.

**Cell apoptosis assay**. The cellular apoptosis rate was determined using the Annexin V/PI double-staining Kit (Dojindo Laboratories, AD10). Cells were seeded in 6-well culture plates at a density of $2.0 \times 10^5$ cells/well. After incubation with different concentrations of L-sorbose for 24 h, the cells were washed with annexin V binding buffer and double-stained with annexin V and PI. Cell apoptosis was examined on a FACSCalibur flow cytometer (BD Accuri C6) and analyzed by FlowJo software.

**Measurement of ROS and mitochondrial volume**. The Intracellular ROS levels were detected by DCFH-DA (Beyotime Biotechnology, S0033S). The Intracellular mitochondrial ROS levels were detected by Dihydrorhodamine 123 (MCE, HY-101894). Cells were cultured in six-well plates at a density of $2.0 \times 10^5$ cells/well. After being treated with L-sorbose, the cells were gently washed with D-PBS (BBI, E607009) followed by incubation with DCFH-DA or Dihydrorhodamine 123 at 37 °C for 30 min.

To obtain microscopy images, cells were seeded on a confocal dish overnight and treated with L-sorbose for 6 h. After being treated with L-sorbose, the cells were gently washed with D-PBS followed by incubation with MitoBright LT (Dojindo Laboratories, MT11) and DCFH-DA (Dojindo Laboratories, CK04) at 37 °C for 30 min, or Dihydrorhodamine 123 alone. Then the cells were washed with D-PBS twice. Images were obtained with a Nikon C2 Eclipse Ti-E inverted confocal microscope equipped with NIS-Element AR software.

To measure the volume of mitochondria, cells were cultured in six-well plates at a density of $2.0 \times 10^5$ cells/well. After being treated with L-sorbose, the cells were gently washed with D-PBS followed by incubation with MitoBright at 37 °C for 30 min. Then the cells were washed twice with D-PBS. The fluorescence of the cells was measured immediately on a FACSCalibur flow cytometer and analyzed by FlowJo software.

**Measurement of mitochondrial membrane potential**. Cells were cultured in 6-well plates at a density of $2.0 \times 10^5$ cells/well. After being treated with L-sorbose, the cells were gently washed with D-PBS. The changes in mitochondrial membrane potential were detected by staining with JC-1 dye (Beyotime Biotechnology, C2006). After incubating with 10 μM JC-1 staining solution in a cell incubator at 37 °C for 15 min, the cells were washed twice with JC-1 staining buffer and then analyzed by flow cytometry.

**Immunofluorescence assay**. Cells were fixed in cold 4% PFA for 20 min and 1% Triton X-100 for 15 min. After being washed twice in 1× PBS for 5 min, cells were incubated in 1% BSA for 30 min. Then being washed in 1× PBS and incubated in 1% BSA containing primary antibodies for 30 min at 37 °C. Cells were washed three times in 1× PBS for 5 min and incubated in 1% BSA containing second antibodies for 30 min at 37 °C. Then being washed and incubated in hoechst33342 for 30 min. Images were obtained with a Nikon C2 Eclipse Ti-E inverted confocal microscope equipped with NIS-Element AR software.

**Intracellular HK activity measurement**. Cells were seeded into six-well dishes at a density of $2.0 \times 10^5$ cells/well. Cells were treated with L-sorbose for 24 h and then collected for the measurement of HK activity using a Hexokinase Activity

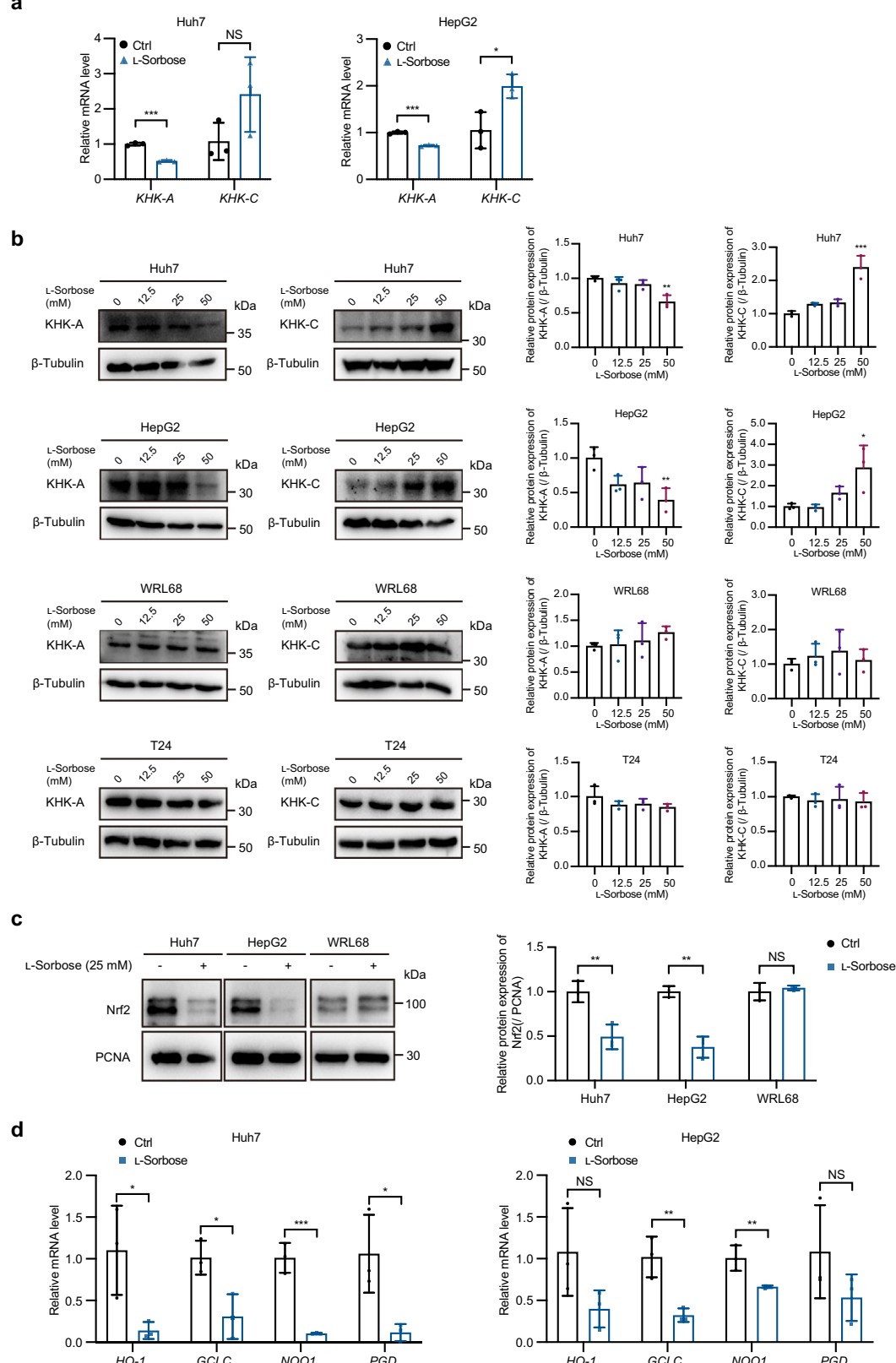

Detection Kit (Solarbio, BC0740) according to the manufacturer's protocol. HK activity was calculated and normalized to the cell number.

**Production of recombinant ketohexokinase**. The plasmid pET28a-*KHK* was transformed into *Escherichia coli* BL21 cells. BL21 cells were grown to an absorbance of 0.6-0.8 at 600 nm and then induced with 0.1 mM IPTG for 20 h at 16 °C.

Cells were lysed in lysis buffer (50 mM Tris-HCl pH 7.5), and the cell lysate was precipitated by centrifugation. The enzyme was purified by batch binding to Ni-NTA resin (Qiagen, 30721). The resin was then washed with lysis buffer containing 250 mM imidazole, and 2His-tagged KHK was eluted with 500 mM imidazole. The purified enzyme was concentrated and desalted with an Amicon Ultra centrifugal filter (10 kDa) using 50 mM Tris-HCl (pH 7.5). The amounts of purified protein were determined by BCA protein assay kit (Beyotime Biotechnology, P0011).

**Fig. 8 L-Sorbose treatment downregulates the expression of KHK-A and inactivates Nrf2-regulated antioxidant defense. a** The mRNA levels of KHK-A and KHK-C in cells with or without (ctrl) 25 mM L-sorbose treatment for 24 h. $n = 3$. **b** Left panels: Western blotting was performed to analyze KHK-A and KHK-C expression in cells after L-sorbose treatment for 24 h. Right panels: Relative intensities of KHK-A and KHK-C. The intensity of KHK-A and KHK-C normalized to β-tubulin detected in cells without L-sorbose treatment were defined as 1. $n = 3$. **c** Left panels: Western blot of Nrf2 in the nuclear fractions of Huh7, HepG2 and WRL68 cells treated with or without (ctrl) 25 mM L-sorbose for 24 h. Right panels: Relative intensities of Nrf2. The intensity of Nrf2 normalized to PCNA detected in cells without L-sorbose treatment was defined as 1. $n = 3$. **d** Levels of mRNA expression of Nrf2-regulated antioxidative genes in Huh7 and HepG2 cells treated with or without (ctrl) 25 mM L-sorbose for 24 h. The mRNA level detected in cells without L-sorbose treatment was defined as 1, and relative mRNA levels detected in L-sorbose-treated cells are shown. $n = 3$. Data are presented as the mean ± s.d. and were analyzed by unpaired two-tailed Student's t test. $*P < 0.05$, $**P < 0.01$, $***P < 0.001$. NS not significant.

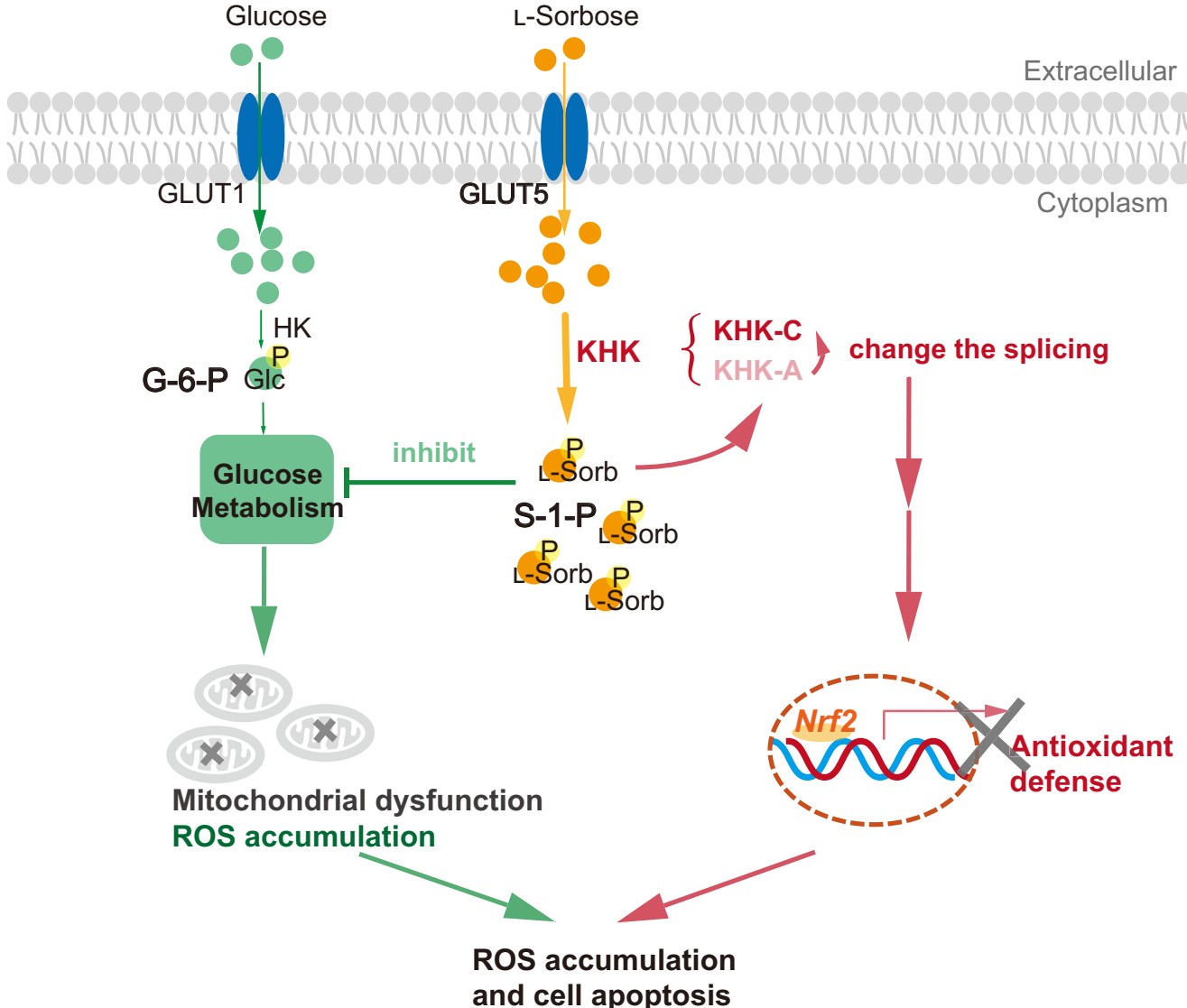

**Fig. 9 Model of the antitumor mechanism of L-sorbose.** L-Sorbose enters cells primarily through GLUT5 and is then converted to L-sorbose-1-phosphate (S-1-P) by KHK. S-1-P inhibits the activity of HK, which induces mitochondrial ROS production and apoptotic cell death. Furthermore, S-1-P downregulates the expression of KHK-A by modulating the splicing mechanism, which results in attenuation of the Nrf2 antioxidation pathway.

**Thin-layer chromatography (TLC) analysis**. The samples were spotted at the bottom of the plate (~1.5 cm above the edge), and the TLC plates (Merck, 1057350001) were developed in a vertical developing chamber with the developing agent glacial acetic acid:isopropanol:water = 2:2:1. After development, the spots were detected by spraying 5% anisaldehyde solution and heat to color development.

**In vitro enzyme inhibition assay**. Hexokinase activity was assayed through a coupled reaction with glucose-6-phosphate dehydrogenase (G6PDH) followed by NADP + detected at 340 nm. Briefly, 10 μL recombinant HK2 (1 μM) and 10 μL S-1-P were incubated together at 37 °C for 10 min. Then 80 μL assay mix

containing 100 mM Tris HCl pH 8.0, 200 mM Glucose, 5 mM MgCl$_2$, 0.8 mM ATP, 1 mM NAD+, 0.25 Units of G6P-DH, was added. Different S-1-P concentrations were added to the incubation mixture described above to investigate the inhibitory effects.

**Statistics and reproducibility**. GraphPad Prism8 was used for statistical analysis. At least three independent or parallel experiments were performed for statistical analysis. Data were analyzed by the unpaired Student's t test, one-way ANOVA with Dunnett's test, or two-way ANOVA with Dunnett's test. Three levels of significance were determined: $*P < 0.05$, $**P < 0.01$, $***P < 0.001$.

**Reporting summary**. Further information on research design is available in the Nature Portfolio Reporting Summary linked to this article.

## Data availability

The data supporting the findings of this study are available within the Article and its Supplementary Information. Source data of the graphs presented in the figures are available in Supplementary Data 1. Unedited immunoblots are provided as Supplementary Figs. 9 and 10 in the Supplementary Information file. Further relevant data are available from corresponding authors upon reasonable request.

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

## Acknowledgements

We are grateful to members of the Gao laboratory for reagents, comments and other contributions to this project. This work was supported by grants-in-aid from the National Natural Science Foundation of China (31971216; 32071467; 32171475; 32101031), Natural Science Foundation of Jiangsu Province (No. BK20210465), Collaborative Innovation Center of Jiangsu Modern Industrial Fermentation, Top-notch Academic Programs Project of Jiangsu Higher Education Institutions, National first-class discipline program of Light Industry Technology and Engineering (LITE2018-015), Shandong Provincial Major Scientific and Technological Innovation Project (2019JZZY011006), Special fund for Zaozhuang Excellence agglomeration project to X-D Gao.

## Author contributions

Conceptualization, X.-D.G., Z.L., H.N., and H.-L.X.; methodology, H.-L.X., X.Z., S.C., and Z.L.; validation, H.-L.X. and S.X.; formal analysis, H.-L.X., X.Z., S.C., and H.N.; investigation, H.-L.X., X.-D.G., and Z.L.; writing—original draft preparation, H.-L.X. and H.N.; writing—review and editing, H.N., H.-L.X., Z.L., and X.-D.G.; supervision, X.-D.G., H.N., and Z.L.; funding acquisition, X.-D.G. and Z.L. All authors have read and agreed to the published version of the manuscript.

## Competing interests

The authors declare no competing interests.
