## [Peer Review file · Communications Biology]

Rare sugar L-sorbose exerts antitumor activity by impairing glucose metabolismReviewers' comments:

Reviewer #1 (Remarks to the Author):

Xu et al. described the novel antitumour effects of L-sorbose. The authors showed, unlike mannose, L-sorbose triggers apoptosis through combined inhibition of HK2 activity via upregulation of KHK-C and impairment of ROS defense via down-regulation of KHK-A. They used mouse xenograft to show L-sorbose worked synergistically with sorafenib, and produced similar results using other antitumour drugs in vitro. What's exciting is cell line sensitivity to L-sorbose correlated with KHK levels, in particular the overexpression of KHK-C isoform increased sensitivity to L-sorbose. Overall, the results are convincing and have substantiated authors conclusion. The work is original and will be of significant interest.

There are still a few major unresolved questions that could bolster the work performed by the authors.

1. Please clarify (line 168), for cell lines that are insensitive to L-sorbose, whether they cannot increase expression of KHK-C in the presence of L-sorbose, or simply have lower baseline expression of KHK-C (Fig. 6a).
2. Please provide flux data (something like in Gonzalez et. al. 2018) to verify to what extent L-sorbose inhibited glycolysis (Line 132). For example, elevated ROS, loss of glucose-6P, accumulation of S-1-P, change in KHK-A vs KHK-C expression, etc?
3. Line 134. In galactose, does L-sorbose induce the same pattern of downregulation of KHK-A and upregulation of KHK-C? What was ROS levels? This can be important to tease apart the dominant arm cytotoxic effects were achieved, through HK inhibition or KHK-A downregulation.
4. Mouse xenograph (in vivo) results lacked matching ROS or expression data to validate cytotoxic effects observed in cell cultures (in vitro).
5. Line 211. There was no direct evidence (enzyme assay) to show S-1-P inhibited HK2 activity, apart from a reference to a 1950 paper.

Minor:

1. Please comment through what mechanism L-sorbose bias KHK splice variants from KHK-A to KHK-C (Line 172, 186)?
2. Fig 4d. Did HK2 activity change due to changes in phosphorylation state and localization?
3. The statement "L-sorbose is harmless in human" ought to be carefully reviewed. Briefly searching, I couldn't find any information if L-sorbose at high dose (mM quantities) is well-tolerated. Unlike L-sorbose, mannose is abundant in natural food and used as supplements.
4. Fig 4a. It is not clear what cells were used to generate the metabolomic data.

Reviewer #2 (Remarks to the Author):

This manuscript describes anti-tumor activity of L-sorbose, a rare sugar that may exert anti-tumor activity. Using a variety of transformed cell lines, the authors demonstrate that L-sorbose decreased cell viability and increases apoptotic cells in culture and in a xenograft model. The authors go on to show that phosphorylation of L-sorbose at the 1 position is required for cell death. The authors attempt to demonstrate that cell death is due to impairing glucose metabolism. This mechanism is not firmly established by the data shown. There are several questions to be answered to help fully interpret the data shown.

1. Figures 1, 2 and 3 focus on establishing the conditions under which L-sorbose causes cell death in several cancer-derived cell lines. In several instances, the variability in the data is so great and the number of independent samples is so low, that it is hard to determine if a difference between control and experimental conditions exists or not. For examples, please see figure 3C and E. Importantly, the variability in the tumor weight of the control cells is so great that it is hard to know if L-sorbose really

makes a difference. There is significant overlap in weight of 2/3 of the of the tumor samples between control and L-sorbose treated mice.

2. The number of cell lines that are used is confusing because the results are similar, yet there are differences that all into the question the conclusions. For example, figure 2 is devoted to mitochondrial function and correlating mitochondrial membrane potential to ROS production. The relationship between membrane potential and ROS production in the two cell types was very different, as was the time course. In figure 2A and B, why are the y-axis not all labeled the same?

3. The metabolomics presented in Figure 4 are hard to interpret. What is the rationale for presenting these particular metabolites. It is not clear how the peaks were identified. Also, it is unclear if a standard was used to correct for efficiency of extraction. The variability is quite high in the control cells, and this makes interpretation very difficult.

4. It is unclear if the authors are concluding that L-sorbose is inhibiting hexokinase activity via an allosteric mechanism or through some other mechanism. No evidence is presented to indicate that this is the case. The conclusion that the HK protein expression is unchanged by L-sorbose treatment is based on data with very high variability. No conclusion can be drawn from such variable data.

5. The authors show that treatment with galactose prevented the L-sorbose effect because galactose enters glycolysis independent of HK. This conclusion is too strong. Galactose is not metabolized well by many cancer cell types, and itself is thought to interfere with glucose metabolism. The way that the data are presented in figure 4 C, it is unclear if galactose itself was inhibited viability, and L-sorbose did not add to that existing effect.

6. The competition with fructose is difficult to interpret without understanding what is used to balance the osmolarity of the sugar treatments. Transport assays would help to clarify if transport, phosphorylation or both are mediating the effects.

7. The clearest data come from the KHK-C knockout and overexpression experiments. These data strongly support that L-sorbose needs to phosphorylated to induce cell death. That said, the correlation between KHK-C expression in the various cell lines and cell viability do not closely correlate (Figure 6A and B). In particular the variability in protein expression for the A549 cell line is confusing. One value is very high.

8. Figure 7 brings in a new cell type (WRL68) to use as a control for L-sorbose-dependent changes in KHK-A and KHK-C expression. The authors have not shown that WRL68 cells are not killed by L-sorbose.

9. Experiments with L-sorbose and Sorafenib 2 by 2 block design. This should be analyzed using 2-way ANOVA to determine interactions between the two compounds.

Reviewer #3 (Remarks to the Author):

In the manuscript submitted, Xu and colleagues describe the role of L-sorbose as a potential antitumor agent. Briefly, L-sorbose is internalized via GLUT5 and inhibits the glycolytic enzyme hexokinase. Notably, this sugar downregulates the hexokinase isoform A expression explicitly, which increases oxidative stress levels and induces apoptosis. The story is exciting, encouraging, and intensely interesting in the scientific community. However, there are several issues to address before publication. The quality of western needs to be highly considered as well as a better metabolic characterization of the tumor cells after L-sorbose. I detail below the specific points to address:

1- The quality of some representative western blots is unacceptable for publication: Figure 4d

(problem of transfer?), Figure 6a,c,f (issue of migration), figure 7a,d (transfer again). Taking into account your good detection, please use other representative blots. In Fig 5b, the loading control is lower in KO cells. Is it always the case? Why is actin used as a loading control in some blots and others tubulin or PCNA? Please, use the same loading control or justify using three different proteins.

2- Although the results are convincing, I consider it better to address the cells' metabolic profile after sorbose treatment. I hardly recommend monitoring the glycolytic and mitochondrial activity with the Seahorse technology (or similar). In addition, not all tumor cells are glucose-sensitive (for example, they use glutamine or lactate to feed the TCA cycle). Did you consider evaluating the effects of L-sorbose depending on the nutrient requirements of tumor cells?

3- The in vivo experiments with the xenograft models are outstanding. However, I miss monitoring glucose/insulin levels in plasma to see whether L-sorbose treatment could impact glycemia levels.

4- In the manuscript, you consider DCFDA a dye monitoring mitochondrial ROS. However, it is a general ROS dye despite having an almost complete colocalization with mitochondria (Fig 2g). Thus, I suggest using another mitochondria dye like Dihydrorhodamine 123.

5- For all the data using FACS, please show a representative histogram. In Fig 2h, I miss the MFI values.

6- Instead of talking about mitochondrial volume, it is more accurate to speak of mitochondrial biomass.

Response to referees

Thank you very much for your valuable comments on our manuscript entitled “Rare sugar L-sorbose exerts antitumor activity by impairing glucose metabolism” submitted to *Communications Biology* (Manuscript ID: COMMSBIO-22-2041-T). Our changes including the modifications and corrections requested from reviewers are marked in the revised manuscript. To a better presentation of our results, we have made major rearrangements in the “result” section including figures. But the abstract, introduction and discussion sections remain the same with the original version.

Main changes are listed below:

1. In the original version, the “result” section includes 5 subtitles. They are: (1) L-Sorbose triggers mitochondrial dysfunction and enhances ROS accumulation in cancer cells to induce cell death; (2) The combination of sorafenib with L-sorbose enhances cell death; (3) L-Sorbose interferes with glucose metabolism by targeting hexokinase; (4) The expression level of KHK dictates L-sorbose sensitivity; (5) L-Sorbose regulates the expression of the two KHK isoforms. We have re-arranged and modified them into 6 subtitles. They are: (1) L-Sorbose triggers mitochondrial apoptosis; (2) L-Sorbose enhances the anticancer effect of sorafenib; (3) L-Sorbose internalizes cell through GLUT5, and accumulates as L-sorbose-1-phosphate; (4) The expression level of KHK dictates L-sorbose sensitivity; (5) L-Sorbose-1-phosphate interferes with glucose metabolism by targeting the hexokinase activity; (6) L-Sorbose treatment inactivates Nrf2-regulated antioxidant defense.
2. Figures also have been modified and re-arranged. Mainly, Figure 3 in the original version was divided into two parts and became Figures 3 and 4 in the revised manuscript; The original Figure 4 was modified and changed its order to Figure 7; Contents in original Figures 1, 2, 5 and 6 were modified and re-arranged but their order remains the same. The original Figure 7 was changed to Figure 8 in the revised manuscript.

Our point by point responses to reviewer’s comments are listed below:

Reviewer #1 (Remarks to the Author):

Xu et al. described the novel antitumour effects of L-sorbose. The authors showed, unlike mannose, l-sorbose triggers apoptosis through combined inhibition of HK2 activity via upregulation of KHK-C and impairment of ROS defense via down-regulation of KHK-A. They used mouse xenograft to show L-sorbose worked synergistically with sorafenib, and produced similar results using other antitumour drugs in vitro. What’s exciting is cell line sensitivity to L-sorbose correlated with KHK levels, in particular the overexpression of KHK-C isoform increased sensitivity to L-sorbose. Overall, the results are convincing and have substantiated authors conclusion. The work is original and will be of significant interest. There are still a few major unresolved questions that could bolster the work performed by the authors.

We thank this reviewer for the positive and constructive comments on our study.

Reviewer's comments 1:

Please clarify (line 168), for cell lines that are insensitive to L-sorbose, whether they cannot increase expression of KHK-C in the presence of L-sorbose, or simply have lower baseline expression of KHK-C (Fig. 6a).

Our response:

We thank the reviewer for this valuable suggestion. The T24 cell line had a lower baseline expression of KHK-C than other cell lines (Fig. 6a (Fig. 6e in the revised version)). L-Sorbose treatment did not have much effect on the cell viability in T24 cells (Fig. 6b (Fig. 6f in the revised version)), which means that it is insensitive to L-sorbose. We performed the western blot assay and confirmed that L-sorbose treatment on T24 cells did not influence the expression of KHK-C and KHK-A, indicating a lower baseline expression as the reason for insensitivity. We added the description in the revised manuscript (Page 9; lines 207-208), and the data was presented in Fig. 8b.

Reviewer's comments 2:

Please provide flux data (something like in Gonzalez et. al. 2018) to verify to what extent L-sorbose inhibited glycolysis (Line 132). For example, elevated ROS, loss of glucose-6P, accumulation of S-1-P, change in KHK-A vs KHK-C expression, etc?

Our response:

We thank the reviewer for this valuable suggestion. According to the reviewer's suggestion, we performed the flux experiment and added in Supplementary Fig. 3. Glucose-6-phosphate in two samples of the L-sorbose groups had a decrease compared with the ctrl groups. The flux of the final metabolite lactic acid also had a decrease in L-sorbose-treated cells. These results suggested the possibility of a decrease in HK activity by L-sorbose treatment. Meanwhile, we also confirmed that S-1-P directly inhibited the activity of a recombinant HK2 (Fig. 7g). Therefore, L-sorbose treatment does have an effect on glycolysis through the inhibition of HK2 activity. The result has been described in the revised manuscript (Page 8; lines 183-186).

Reviewer's comments 3:

Line 134. In galactose, does L-sorbose induce the same pattern of downregulation of KHK-A and upregulation of KHK-C? What was ROS levels? This can be important to tease apart the dominant arm cytotoxic effects were achieved, through HK inhibition or KHK-A downregulation.

Our response:

We thank the reviewer for this valuable suggestion. We cultured cells in glucose-free DMEM supplemented with galactose and treated them with L-sorbose, and detected the ROS levels and expression of KHK-A and KHK-C. Our result indicated that L-sorbose treatment did not change the ROS level, nor the expression of KHK-A and KHK-C. The data have been added in Fig. 7d and Supplementary Fig. 6. Compared with the changes in the glucose culture medium, we make a conclusion that the elevated ROS level and the downregulation expression of KHK-A and the

upregulation of KHK-C both achieve the antitumor effect of L-sorbose. The related results have been described in the revised manuscript (Page 9; lines 192-193 and Page 9; lines 207-209).

Reviewer's comments 4:

Mouse xenograph (in vivo) results lacked matching ROS or expression data to validate cytotoxic effects observed in cell cultures (in vitro).

Our response:

We thank the reviewer for pointing it out. We performed western blot and IHC assay to detect the expression of HK2 in the mice tumor tissues. The result indicated that HK2 expression did not change after L-sorbose treatment (Fig. 7f and Supplementary Fig. 5b). In addition, the ratio of BAX/Bcl2 increased in the sorafenib combined with L-sorbose groups, revealing the phenotype of apoptosis (Fig. 4d). The dihydrorhodamine 123 was used to monitor ROS. Our result indicated that the ROS level was evaluated in the sorafenib combined with L-sorbose groups (Fig. 4e). We added the description in the revised manuscript (Page 7; lines 132-134).

Reviewer's comments 5:

Line 211. There was no direct evidence (enzyme assay) to show S-1-P inhibited HK2 activity, apart from a reference to a 1950 paper.

Our response:

We performed *in vitro* assay to show direct inhibition of S-1-P to HK2 activity. We used pure enzyme HK2 and different concentrations of S-1-P compound co-incubation to measure the inhibition rate. As shown in Fig. 7g, S-1-P inhibited HK2 activity with 4.35 mg/mL of IC50. We added the description in the revised manuscript (Page 9; lines 199-201).

Minor:

Reviewer's comments 1:

Please comment through what mechanism L-sorbose bias KHK splice variants from KHK-A to KHK-C (Line 172, 186)?

Our response:

Our results showed that it's not L-sorbose but the S-1-P directly affects the splicing of KHK. It is reported that hnRNPH1/2 blocks the recognition of the exon 3C 5' splice site of KHK pre-mRNA, preventing recognition of exon 3C and favoring the inclusion of exon 3A. S-1-P may be combined with hnRNPH1/2 in cells and influences the binding of pre-mRNA with hnRNPH1/2 to promote a switch from KHK-A to KHK-C expression. We have discussed this possible mechanism in the revised manuscript (Page 12; lines 263-265).

Reviewer's comments 2:

In Fig 4d, Did HK2 activity change due to changes in phosphorylation state and localization?

Our response:

We performed the anti-HK2 immunofluorescence assay and demonstrated that the location of HK2 was not changed in the cells with L-sorbose treatment (Supplementary Fig. 5a). In addition, IP and WB assays revealed that the phosphorylation of HK2 had no change in cells with L-sorbose treatment (Supplementary Fig. 5c). We added the description in the revised manuscript (Page 9; lines 197-199).

Reviewer's comments 3:

The statement “L-sorbose is harmless in human” ought to be carefully reviewed. Briefly searching, I couldn't find any information if L-sorbose at high dose (mM quantities) is well-tolerated. Unlike L-sorbose, mannose is abundant in natural food and used as supplements.

Our response:

We totally agree with reviewer's point. It has been shown that L-sorbose did not induce hemolysis of human erythrocytes¹. Meanwhile, L-sorbose could be metabolized in the rat without toxicity^{2,3}. In our xenograft model, L-sorbose had less influence on mouse weight. However, it still needs further study to show whether it is harmless in humans at high doses. Following the advice from reviewer, we deleted this sentence. The related information has been discussed in the manuscript (Page 12; lines 270-273).

1. Kistler A, Keller P. Inhibition of glycolysis by L-sorbose in dog erythrocytes. *Experientia*. **34**, 800 (1978).
2. Würsch P, Welsch C, Arnaud MJ. Metabolism of L-sorbose in the rat and the effect of the intestinal microflora on its utilization both in the rat and in the human. *Nutr Metab*. **23**, 145-155 (1979).
3. BURNS JJ, MOSBACH EH, SCHULENBERG S, REICHENTHAL J. Studies on the incorporation of C14 administered as L-sorbose into L-ascorbic acid and D-glucose in rats. *J Biol Chem*. **214**, 507-514 (1955).

Reviewer's comments 4:

Fig 4a. It is not clear what cells were used to generate the metabolomic data.

Our response:

We thank this reviewer for pointing this out. The Huh7 cells were used in the metabolomic assay. It has been presented in the manuscript (Page 8; line 178; Fig. 7a).

Reviewer #2 (Remarks to the Author):

This manuscript describes anti-tumor activity of L-sorbose, a rare sugar that may exert anti-tumor activity. Using a variety of transformed cell lines, the authors demonstrate that L-sorbose decreased cell viability and increases apoptotic cells in culture and in a xenograft model. The authors go on to show that phosphorylation of L-sorbose at the 1 position is required for cell death. The authors attempt to demonstrate that cell death is due to impairing

glucose metabolism. This mechanism is not firmly established by the data shown. There are several questions to be answered to help fully interpret the data shown.

We thank this reviewer for your precious advice for our study.

Reviewer's comments 1:

Figures 1, 2 and 3 focus on establishing the conditions under which L-sorbose causes cell death in several cancer-derived cell lines. In several instances, the variability in the data is so great and the number of independent samples is so low, that it is hard to determine if a difference between control and experimental conditions exists or not. For examples, please see figure 3C and E. Importantly, the variability in the tumor weight of the control cells is so great that it is hard to know if L-sorbose really makes a difference. There is significant overlap in weight of 2/3 of the of the tumor samples between control and L-sorbose treated mice.

Our response:

We thank the reviewer for pointing this out. Several results need to be improved and more rigorously analyzed. We have repeated the animal experiment in $n = 8$ per group and replaced the figures. Data in this result with more animals and lower variability suggested a greater inhibitory effect on tumor growth when sorafenib was administered in combination with L-sorbose in mouse xenograft models. L-Sorbose does make a difference. Related data can be found in Figs. 4a, b, c, f and Supplementary Fig. 2 in the revised version. Also, to improve the results in Fig. 3E (Fig. 3c in the revised version), we added other three independent experiments and made an analysis in total. It obtained a similar result showing that L-sorbose- and sorafenib-treated cells died due to the induction of apoptosis and cell death induced by sorafenib and L-sorbose was alleviated by incubation with the pancaspase inhibitor z-VAD-FMK.

Reviewer's comments 2:

The number of cell lines that are used is confusing because the results are similar, yet there are differences that all into the question the conclusions. For example, figure 2 is devoted to mitochondrial function and correlating mitochondrial membrane potential to ROS production. The relationship between membrane potential and ROS production in the two cell types was very different, as was the time course.

Our response:

We thank the reviewer for pointing this out. We chose two cell lines Huh7 and HepG2 for further study because they both were liver cancer cell lines. Although they have some differences in the change levels of MMP and ROS after L-sorbose treatment, the trend of changes compared with the control group was same.

In Figs. 2a and b, although Huh7 cell line was more sensitive with L-sorbose treatment, both Huh7 and HepG2 cells revealed a decrease in mitochondrial membrane potential (MMP) after L-sorbose treatment indicating the dysfunction of mitochondria. Similar in Figs. 2c and d, ROS levels were significantly elevated when the two cells were treated with 50 mM L-sorbose for 24 h and 25 mM L-sorbose for 6 h and 9 h. The difference is that, unlike HepG2 cells, Huh7 cells showed a

decreased ROS level after 12 h treatment. However, the decrease in cell viability has been observed from 6 h, indicating the inhibition of L-sorbose occurred before (Fig. 2e).

We also used dihydrorhodamine 123 to measure the mitochondrial ROS level in these two cancer cells. The result showed that both Huh7 and HepG2 cell lines had an increase in L-sorbose-treated cells (Supplementary Fig. 1a). These results suggested that the changes in MMP and ROS production in cells after L-sorbose treatment should have contributed to the effect of cell apoptosis.

Reviewer's comments 2:

In figure 2A and B, why are the y-axis not all labeled the same?

Our response:

We apologize for our careless mistake. The correction has been made in Figs. 2a and b in the revised version.

Reviewer's comments 3:

The metabolomics presented in Figure 4 are hard to interpret. What is the rationale for presenting these particular metabolites. It is not clear how the peaks were identified. Also, it is unclear if a standard was used to correct for efficiency of extraction. The variability is quite high in the control cells, and this makes interpretation very difficult.

Our response:

We performed central carbon metabolomic analyses including glycolysis, pentose phosphate pathway and TCA cycle. The reason is that L-sorbose might interfere with glucose metabolism.

The standard solution of the target compound was injected into the mass spectrum before HPLC-MS/MS analysis. For each target compound, select several ion pairs with the highest signal strength, optimize their MRM parameters and select the ion pair with the best response for quantitative analysis and other ion pairs for qualitative analysis of the target compound.

Reviewer's comments 4:

It is unclear if the authors are concluding that L-sorbose is inhibiting hexokinase activity via an allosteric mechanism or through some other mechanism. No evidence is presented to indicate that this is the case. The conclusion that the HK protein expression is unchanged by L-sorbose treatment is based on data with very high variability. No conclusion can be drawn from such variable data.

Our response:

We thank the reviewer for pointing this out. Our results showed that it's not L-sorbose but the S-1-P induce cell death. In the response to reviewer 1, we have described an in vitro enzyme inhibition assay, which showed that S-1-P directly inhibited the activity of HK2 in a dose-dependent manner. Meanwhile, there were no changes found in the expression, location and phosphorylation of HK2 (Fig. 5a; Figs. 7e, f; Supplementary Figs. 5b, c). These results demonstrated that S-1-P inhibits HK2 activity via an allosteric mechanism.

Reviewer's comments 5:

The authors show that treatment with galactose prevented the L-sorbose effect because galactose enters glycolysis independent of HK. This conclusion is too strong. Galactose is not metabolized well by many cancer cell types, and itself is thought to interfere with glucose metabolism. The way that the data are presented in figure 4 C, it is unclear if galactose itself was inhibited viability, and L-sorbose did not add to that existing effect.

Our response:

We thank the reviewer for this valuable suggestion. To answer whether galactose itself inhibits cell viability, we performed the CCK-8 assay to detect the cell viability in Huh7 and HepG2 cells with 0-100 mM concentrations of D-galactose, and we found that galactose itself does not inhibit viability (Supplementary Fig. 4). In addition, the ROS level was also analyzed by DCFH-DA in Huh7 cells cultured in glucose-free DMEM supplemented with D-galactose after 25 mM L-sorbose treatment at different time points (Fig. 7d). There were no changes in ROS level compared to the ctrl group, suggesting galactose does not trigger cell death.

Reviewer's comments 6:

The competition with fructose is difficult to interpret without understanding what is used to balance the osmolarity of the sugar treatments. Transport assays would help to clarify if transport, phosphorylation or both are mediating the effects.

Our response:

We thank the reviewer for pointing this out. We agree that it lacks explanation on the fructose inhibition experiment leading the difficulty in interpretation. This experiment was designed to test whether L-sorbose can compete the enzymes in fructose metabolism pathway because of its structural similarity with D-fructose. Based on this experiment, our knockout experiments have confirmed that L-sorbose entered cells through GLUT5 and phosphorylated by KHK. However, in this study, we focused on the anti-tumor effect of L-sorbose treatment under the glucose condition. There is no competition between L-sorbose and D-fructose under the glucose condition. If only focusing on the fructose inhibition experiment, as reviewer indicated, the competitive effects of fructose including osmolarity ought to be considered. Even though, we found that the high concentration (100 mM) of fructose did not affect the growth of Huh7 and HepG2 cells.

Reviewer's comments 7:

The clearest data come from the KHK-C knockout and overexpression experiments. These data strongly support that L-sorbose needs to phosphorylated to induce cell death. That said, the correlation between KHK-C expression in the various cell lines and cell viability do not closely correlate Figure 6A and B). In particular the variability in protein expression for the A549 cell line is confusing. One value is very high.

Our response:

We thank the reviewer for pointing this out. The difference was significant after overexpression of KHK-C in cells, suggesting that it is a critical enzyme to dictate sensitivity to L-sorbose (Figs. 6a, b). In consistent with this founding, A549 cell line with the highest expression of KHK-C among these 7 cell lines has the largest response on cell viability after L-sorbose treatment. While T24 cell

line with low KHK-C expression was not much sensitive to L-sorbose. Thus, we think that L-sorbose sensitivity was correlated with the levels of KHK-C. To make the results more reliable, we have repeated the experiment of the KHK-C expression in seven cell lines. The new results have been presented in Fig. 6e.

Reviewer's comments 8:

Figure 7 brings in a new cell type (WRL68) to use as a control for l-sorbose-dependent changes in KHK-A and KHK-C expression. The authors have not shown that WRL68 cells are not killed by L-sorbose.

Our response:

Following the suggestion from the reviewer, we have detected cell viability of WRL68 cells in different concentrations of L-sorbose. The result showed that it had little influence on cell viability. We presented the result in Supplementary Fig. 7 and added the description in the revised manuscript (Page 10; lines 220).

Reviewer's comments 9:

Experiments with L-sorbose and Sorafenib 2 by 2 block design. This should be analyzed using 2-way ANOVA to determine interactions between the two compounds.

Our response:

We thank the reviewer for pointing this out. We have changed to the 2-way ANOVA to analyze the experiments with L-sorbose and sorafenib in Figs. 3 and 4.

Reviewer #3 (Remarks to the Author):

In the manuscript submitted, Xu and colleagues describe the role of L-sorbose as a potential antitumor agent. Briefly, L-sorbose is internalized via GLUT5 and inhibits the glycolytic enzyme hexokinase. Notably, this sugar downregulates the hexokinase isoform A expression explicitly, which increases oxidative stress levels and induces apoptosis. The story is exciting, encouraging, and intensely interesting in the scientific community. However, there are several issues to address before publication. The quality of western needs to be highly considered as well as a better metabolic characterization of the tumor cells after L-sorbose. I detail below the specific points to address:

We thank this reviewer for the positive and constructive comments on our study.

Reviewer's comments 1:

The quality of some representative western blots is unacceptable for publication: Figure 4d (problem of transfer?), Figure 6a, c, f (issue of migration), figure 7a, d (transfer again). Taking into account your good detection, please use other representative blots. In Fig 5b, the loading control is lower in KO cells. Is it always the case? Why is actin used as a loading

control in some blots and others tubulin or PCNA? Please, use the same loading control or justify using three different proteins.

Our response:

In this study, basically actin was used as the loading control. However, actin is not suitable for analyzing KHK because its molecular weight is close to KHK-A and KHK-C and hard to be separated in the same SDS-PAGE. So, tubulin was used instead. We also used PCNA as the control for analyzing Nrf2 because they both are nuclear proteins.

To improve the quality of the results, we performed new western blot analysis. The results are presented in Fig. 5c, Fig. 6e, Fig. 6j, Fig. 7e, and Fig. 8c.

Reviewer's comments 2:

Although the results are convincing, I consider it better to address the cells' metabolic profile after sorbose treatment. I hardly recommend monitoring the glycolytic and mitochondrial activity with the Seahorse technology (or similar).

Our response:

We thank the reviewer for the suggestion. We may didn't explain well in the manuscript. We did perform the cell metabolic profile after L-sorbose treatment and focused mainly on the central carbon metabolism pathway (Fig. 7a). In addition, following the suggestion from reviewer 1, we further performed the metabolism flux analysis using $^{13}\text{C}_6$ -glucose to track the level of metabolites mainly in glycolysis (Supplementary Fig. 3). Both results demonstrated that L-sorbose inhibits the glucose metabolism in cancer cells.

Reviewer's comments 2:

In addition, not all tumor cells are glucose-sensitive (for example, they use glutamine or lactate to feed the TCA cycle). Did you consider evaluating the effects of L-sorbose depending on the nutrient requirements of tumor cells?

Our response:

We so appreciate the reviewer for this useful suggestion. Effects of L-sorbose treatment on glucose insensitive cells have been discussed in the revised manuscript (Page 12; lines 265-267).

Reviewer's comments 3:

The in vivo experiments with the xenograft models are outstanding. However, I miss monitoring glucose/insulin levels in plasma to see whether L-sorbose treatment could impact glycemia levels.

Our response:

We thank the reviewer for this valuable suggestion. The glucose/insulin levels of mice have been detected. We found that L-sorbose does not impact mice glycemia levels. The results have been presented in Figs. 4g, h and described in the revised manuscript (Page 7; lines 135-136).

Reviewer's comments 4:

In the manuscript, you consider DCFDA a dye monitoring mitochondrial ROS. However, it is a general ROS dye despite having an almost complete colocation with mitochondria (Fig 2g). Thus, I suggest using another mitochondria dye like Dihydrorhodamine 123.

Our response:

We thank the reviewer for this valuable suggestion. We performed the FACS assay using dihydrorhodamine 123 to measure the mitochondrial ROS level in cancer cells and tumor tissues. An increase of ROS level was detected in L-sorbose treatment cells (Supplementary Fig. 1a). In addition, the ROS level was also evaluated in tumor tissues from the mice treated with L-sorbose alone and combination with sorafenib (Fig. 4e). We added the related description in the revised manuscript (Page 6, lines 103-104; Page 7, lines 133-134).

Reviewer's comments 5:

For all the data using FACS, please show a representative histogram. In Fig 2h, I miss the MFI values.

Our response:

The representative histograms or dot plots have been added to all the FACS data in the revised version. The MFI value has been added in Fig. 2h.

Reviewer's comments 6:

Instead of talking about mitochondrial volume, it is more accurate to speak of mitochondrial biomass.

Our response:

We have changed "volume" to "biomass" in the revised manuscript (Page 6; line 105).

Reviewers' comments:

Reviewer #1 (Remarks to the Author):

Authors have made commendable efforts to address my comments.

Minor corrections:

page 6 line 119 shear => share

Please indicate/clarify what data was plotted in Supp Fig 3 - abundance of uniformly labelled metabolites?

Reviewer #2 (Remarks to the Author):

The authors have made a very careful response to the previous critiques. The revised manuscript is significantly improved

Reviewer #3 (Remarks to the Author):

I thank the authors for addressing all the comments from the 3 reviewers. Nice piece of work. Before publication, I suggest improving the brightness of Fig. 2g

Response to Reviewers

Reviewer #1 (Remarks to the Author):

Authors have made commendable efforts to address my comments.

We thank this reviewer for the positive comments on our study.

Minor corrections:

Reviewer's comments 1:

page 6 line 119 shear => share

Our response:

We have changed "shear" to "share" in the revised manuscript (Page 6; line 119).

Reviewer's comments 2:

Please indicate/clarify what data was plotted in Supp Fig 3 - abundance of uniformly labeled metabolites?

Our response:

We use the abundance of uniformly labeled metabolites to make the plot in Supplement Figure 3. It has been indicated in the figure legend of Supplement Figure 3 in the revised manuscript (Page 31; lines 776-777).

Reviewer #3 (Remarks to the Author):

Reviewer's comments:

I thank the authors for addressing all the comments from the 3 reviewers. Nice piece of work. Before publication, I suggest improving the brightness of Fig. 2g

Our response:

We thank this reviewer for the positive and constructive comments on our study. We have improved the brightness with the same adjustment in two groups of Fig. 2g in the revised version for clearer display.